# *Vibrio cholerae*'s mysterious Seventh Pandemic island (VSP-II) encodes novel Zur-regulated zinc starvation genes involved in chemotaxis and cell congregation

**Shannon G. Murphy**[1,2], **Brianna A. Johnson**[1,3], **Camille M. Ledoux**[1,3],
**Tobias Dörr**[1,2,3]*

1 Weill Institute for Cell and Molecular Biology, Cornell University, Ithaca, New York, United States of America, 2 Department of Microbiology, Cornell University, Ithaca, New York, United States of America, 3 Cornell Institute of Host-Microbe Interactions and Disease, Cornell University, Ithaca, New York, United States of America

* tdoerr@cornell.edu

**Data Availability Statement:** Raw RNA-seq files are available from the GEO database (GSE173966, GSE174028). All other relevant data are shown as

## Abstract

*Vibrio cholerae* is the causative agent of cholera, a notorious diarrheal disease that is typically transmitted via contaminated drinking water. The current pandemic agent, the El Tor biotype, has undergone several genetic changes that include horizontal acquisition of two genomic islands (VSP-I and VSP-II). VSP presence strongly correlates with pandemicity; however, the contribution of these islands to *V. cholerae*'s life cycle, particularly the 26-kb VSP-II, remains poorly understood. VSP-II-encoded genes are not expressed under standard laboratory conditions, suggesting that their induction requires an unknown signal from the host or environment. One signal that bacteria encounter under both host and environmental conditions is metal limitation. While studying *V. cholerae*'s zinc-starvation response *in vitro*, we noticed that a mutant constitutively expressing zinc starvation genes (Δ*zur*) congregates at the bottom of a culture tube when grown in a nutrient-poor medium. Using transposon mutagenesis, we found that flagellar motility, chemotaxis, and VSP-II encoded genes were required for congregation. The VSP-II genes encode an AraC-like transcriptional activator (VerA) and a methyl-accepting chemotaxis protein (AerB). Using RNA-seq and *lacZ* transcriptional reporters, we show that VerA is a novel Zur target and an activator of the nearby AerB chemoreceptor. AerB interfaces with the chemotaxis system to drive oxygen-dependent congregation and energy taxis. Importantly, this work suggests a functional link between VSP-II, zinc-starved environments, and energy taxis, yielding insights into the role of VSP-II in a metal-limited host or aquatic reservoir.

## Author summary

The Vibrio Seventh Pandemic island was horizontally acquired by the El Tor pandemic strain, but its role in pathogenicity or environmental persistence is unknown. A major

raw data points within the manuscript and its Supporting Information files.

**Funding:** National Institute of Health grant R01GM130971 to TD. The funders had no role in study design, data collection and analysis, decision to publish, or preparation of the manuscript.

**Competing interests:** The authors have declared that no competing interests exist.

barrier to VSP-II study was the lack of stimuli favoring its expression. We show that zinc starvation induces expression of island components and describe a transcriptional network that activates a VSP-II encoded energy taxis receptor. Importantly, energy taxis may enable *V. cholerae* to locate more favorable microenvironments, possibly to colonize anoxic portions of the gut or environmental sediments.

## Introduction

The Gram-negative bacterium *Vibrio cholerae*, the causative agent of cholera [1], is well-adapted to two distinct lifestyles: as a colonizer of macroinvertebrates in the aquatic environment and as a potentially lethal pathogen inside the human intestine [2]. *V. cholerae* persists in aquatic reservoirs by colonizing a variety of (mostly chitinous) biotic surfaces, such as copepods [3–6], shellfish [7,8], and arthropods [9,10]. *V. cholerae* is ingested via contaminated drinking water or, less commonly, via undercooked seafood [11,12]. Once inside the human host, pathogenic varieties of *V. cholerae* (typically O1 and O139 serovars [13]) rely on virulence factors to establish infection; the toxin co-regulated pilus (TCP) facilitates attachment to the intestinal wall [14,15] and cholera toxin (CTX) secretion ultimately drives efflux of water and salts from the intestinal epithelium [16]. CTX additionally promotes nutrient competition via depletion of free (i.e., not heme-bound) iron in the intestine [17].

The current (seventh) cholera pandemic agent, the O1 serovar El Tor biotype, arose from a non-pathogenic precursor via acquisition of TCP and CTX virulence factors [18]. Unlike its pandemic predecessor (the classical biotype), El Tor underwent several changes that include, among others [19], the development of resistance against the antimicrobial peptide polymyxin B [20,21] and horizontal acquisition of two genomic islands (VSP-I and VSP-II) [22,23]. Presence of VSP-I and -II strongly correlate with pandemicity; however, only genes encoded on VSP-I have been directly linked to increased fitness in a host [24]. VSP-II is a poorly understood 26-kb island that contains 30 annotated ORFs spanning *vc0490-vc0516* [25], only two of which have validated functions: an integrase (*vc0516*, [26]) and a peptidoglycan endopeptidase (*vc0503*, [27]). The remaining uncharacterized genes are predicted to encode transcriptional regulators (VC0497, VC0513), ribonuclease H (VC0498), a type IV pilin (VC0502), a DNA repair protein (VC0510), methyl-accepting chemotaxis proteins (VC0512, VC0514), a cyclic di-GMP phosphodiesterase (VC0515), and hypothetical proteins [23]. It is unclear if or how VSP-II enhances the pathogenicity or environmental fitness of El Tor. Intriguingly, VSP-II genes are not expressed under standard laboratory conditions [28], suggesting that their induction requires an unknown signal from the host or environment.

One signal that bacteria encounter under both host and environmental conditions is metal limitation. Bacteria must acquire divalent zinc cofactors from their surroundings to perform essential cellular processes; however, vertebrate hosts actively sequester zinc and other essential transition metals to limit bacterial growth (i.e. nutritional immunity) [29–32]. In the environment, *V. cholerae* frequently colonize the chitinous exoskeletons of aquatic and marine invertebrates and exposure to chitin oligomers has been suggested to induce zinc and iron starvation in *V. cholerae* [33]. In order to cope with zinc starvation stress, *V. cholerae* encodes a set of genes under the control of the well-conserved Zur repressor. When zinc availability is low, Zur dissociates from a conserved promoter sequence, allowing for transcription of downstream genes. *V. cholerae* genes containing a Zur binding region include those encoding zinc import systems (ZnuABC and ZrgABCDE) [34], ribosomal proteins (RpmE2, RpmJ) [35,36], a GTP cyclohydrolase (RibA) [35], and the VSP-II-encoded peptidoglycan endopeptidase (ShyB) [27].

Here, we show that many genes of the VSP-II island are expressed during zinc starvation. These findings stemmed from an initial observation that a *V. cholerae* Δ*zur* mutant accumulated at the bottom of nutrient-poor liquid cultures. We hypothesized that this behavior was mediated by unidentified members of the Zur regulon. Using a transposon mutagenesis screen and RNA-seq, we identified Zur-regulated congregation factors encoded on VSP-II. These included the transcriptional activator VerA (Vibrio energy taxis regulator A) encoded within the *vc0513-vc0515* operon. VerA activates expression of the AerB (aerotaxis B) chemotaxis receptor encoded by *vc0512*. We show that AerB mediates oxygen-dependent congregation and energy taxis. Importantly, these results implicate a role for VSP-II genes in chemotactic movement during zinc starvation.

## Results

### The V. cholerae Δzur *mutant congregates in minimal medium*

We noticed serendipitously that a *V. cholerae* N16961 Δ*zur* mutant (but not the wild type) accumulated at the bottom of a culture tube when grown shaking overnight in M9 minimal medium (**Fig 1A**). A similar result was observed in static overnight cultures (**S1A Fig**). Microscopic inspection of Δ*zur* cells transferred to an agar pad revealed mostly individual cells with no obvious changes in morphology (**Fig 1B**); the lack of strong cell-to-cell interactions holding these "congregates" together is consistent with the ease at which the pellet was dispersed by agitation (**S1 Movie**). *V. cholerae* aggregations in liquid culture are reportedly mediated by numerous mechanisms (e.g. quorum sensing, attachment pili, and O-antigen synthesis [37–44]) and stimuli (e.g. autoinducers, calcium ions [37], cationic polymers [43]), but none thus far have been tied to zinc homeostasis. We therefore sought to identify factors that were required for Δ*zur* to congregate. Congregation (quantified as the ratio of optical densities ($OD_{600nm}$) in the supernatant before and after vortexing) was alleviated by complementing *zur in trans*, excluding polar effects resulting from *zur* deletion (**Fig 1C**). We next examined the role of zinc availability on congregation. Since metals can absorb to the surface of borosilicate glass culture tubes [45], we instead grew *V. cholerae* in plastic tubes and noted that Δ*zur* still congregated at the bottom (**S1B Fig**), indicating that this phenotype is not linked to the properties of the culture vessel. Imposing zinc starvation via deletion of genes encoding *V. cholerae*'s primary zinc importer ZnuABC caused cells to congregate similarly to the Δ*zur* mutant. Congregation of Δ*znuABC* (which still elaborates the low-affinity zinc transporter ZrgABC [34]) was reversed by zinc supplementation (**Fig 1D**). In contrast, the Δ*zur* mutant, which constitutively expresses zinc starvation genes, congregated in both the presence and absence of exogenous zinc. These data indicate that congregation occurs in minimal medium when the Zur regulon is induced (i.e., during zinc deficiency or in a *zur* deletion strain) and is not a direct consequence of zinc availability *per se*. Surprisingly, none of the annotated members of the Zur regulon were required for congregation (**Figs 1E and S1C**), suggesting that there may be other Zur-regulated congregation genes yet to be identified.

### Δzur *congregation requires motility, chemotaxis, and VSP-II encoded proteins*

We reasoned that we could leverage the Δ*zur* congregation phenotype to identify novel components of *V. cholerae*'s zinc starvation response. To find such Zur-regulated "congregation factors", we subjected the Δ*zur* mutant to transposon mutagenesis and screened for insertions that prevent congregation (see Methods for details) (**Fig 2A**). Δ*zur* transposon libraries were inoculated into M9 minimal medium and repeatedly sub-cultured until no pellet formed.

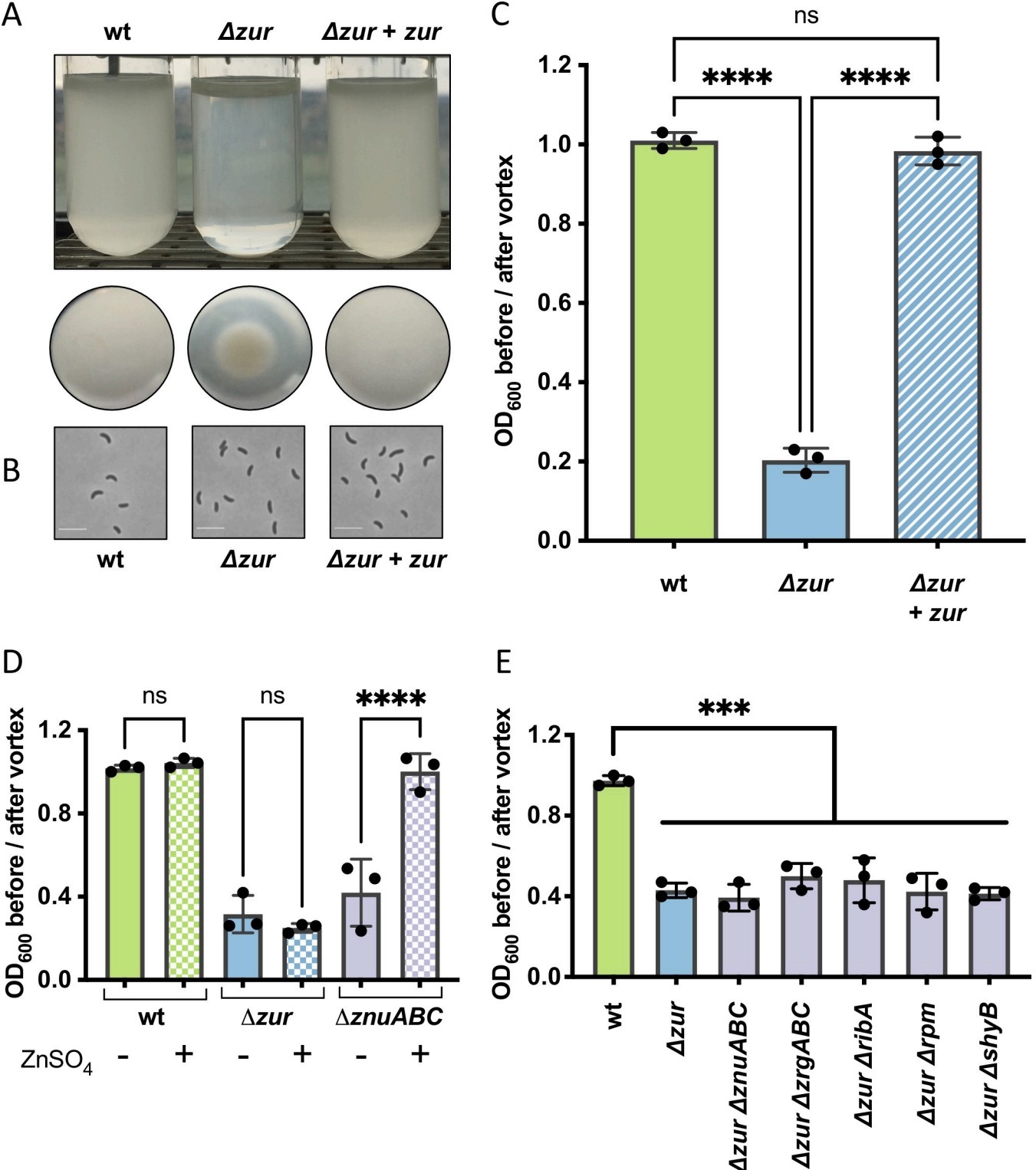

**Fig 1. A *V. cholerae Δzur* mutant congregates in M9 minimal medium.** (**A-C**) Wild-type, *Δzur*, and *Δzur* carrying an integrated, IPTG-inducible copy of *zur* (denoted + *zur*) were grown overnight at 30˚C in M9 minimal medium supplemented with glucose (0.2%) and inducer (IPTG, 500 μM). (**A**) Representative side and bottom-view photos of overnight cultures are shown. (**B**) Cells collected from the bottom of the tube were imaged on an 0.8% agarose pad. Scale bar, 5 μM. (**C**) Congregation was quantified by measuring the optical density (at 600 nm) of the culture supernatant before and after a brief vortex. A ratio close to 1 represents a homogenous culture, a ratio closer to 0 indicates congregation. (**D**) Congregation was measured in wild type, *Δzur*, and a zinc importer mutant

(*ΔznuABC*) grown in M9 glucose (0.2%) in the absence (solid bars) or presence (checkered bars) of exogenous zinc (ZnSO4, 1 μM). (**E**) Congregation in M9 glucose (0.2%) was measured in wild type, *Δzur*, and *Δzur* lacking components of the zinc starvation response (*znuABC*, *zrgABC*, *ribA*, *rpmE2/rpmJ2*, or *shyB*). For all plots, the shown raw data points are biological replicates, error bars represent standard deviation, and asterisks denote statistical difference via Ordinary one-way ANOVA test (\*\*\*\*, $p < 0.0001$; \*\*\*, $p < 0.001$; n.s., not significant).

Transposon insertions sites were identified using arbitrary PCR on isolated colonies [46]. The insertions that prevented congregation overwhelmingly mapped to loci encoding motility and chemotaxis genes (**Fig 2B**). Twenty-four out of 48 recovered transposon mutants were disrupted in flagellar components or motility regulators. We reconstituted these types of mutations in *Δzur* by inactivating flagellum assembly (major flagellin subunit, *fliC*) or rotation (motor protein, *motB*). Both *Δzur ΔfliC* and *Δzur ΔmotB* failed to form a pellet and congregation could be restored by complementing each of these genes *in trans* (**Fig 2C**). These data suggest that *Δzur* congregation is a motility-dependent process. Additionally, seven transposons inserted within genes encoding parts of *V. cholerae's* chemotaxis machinery (*che-2*) (**Fig 2B**); this system modulates bacterial movement in response to a chemical gradient. Mutating a component of this chemotactic phosphorelay (*cheA*::STOP) was sufficient to prevent congregation in *Δzur*, while *trans* expression of *cheA* restored pellet formation to the *Δzur cheA*:: *STOP* mutant (**Fig 2C**). Deletion of other *che-2* open reading frames also prevented *Δzur* from congregating (**S1D Fig**). Collectively, these data suggest that motility and chemotaxis are required for *Δzur* congregation in minimal medium.

We noted that the *Δzur* phenotype resembles aggregation in *E. coli* rough mutants, which have reduced expression of lipopolysaccharides [47]. We observed similar aggregation in *V. cholerae* rough mutants (*vc2205*::*kan*), but this aggregation did not require motility to form and is therefore mediated by a distinct mechanism (**S1E Fig**). We anticipated initially that *Δzur* pellet formation was a group behavior that may require processes associated with surface attachment (e.g., biofilm formation, attachment pili) or cellular communication (e.g., quorum sensing); however, such mutants were not identified by the transposon screen. We thus separately assessed this in a targeted fashion by testing whether *Δzur* still congregates when deficient in biofilm formation *(ΔvspL)* or type IV pili attachment (Δ4: *ΔtcpA ΔmshA ΔpilA*, and orphan pilin *Δvc0502*). Consistent with these processes not answering our screen, biofilm and type IV pili encoding genes were not required for *Δzur* to congregate (**S1F Fig**). Notably, N16961 contains an authentic frameshift mutation in the quorum sensing gene *hapR* [48,49]; however, a repair to *hapR* [50] did not alter congregation dynamics in *Δzur* (**S1G Fig**). We additionally demonstrated that other quorum sensing genes (*ΔcsqA*, *ΔcsqS*, *Δtdh*, *ΔluxQ, or ΔluxS*) were dispensable for this phenotype (**S1F Fig**). Taken together, these data indicate that *Δzur* pellet formation is not a clumping phenomenon driven by typical colonization and congregation factors, but rather a chemotaxis/motility-mediated assembly in the lower strata of a growth medium column.

Since congregation appeared to require induction of the Zur regulon, we were surprised that the transposon screen was not strongly answered by genes with an obvious Zur binding site in their promoters. We reasoned, however, that our screen did not reach saturation due to the large number of motility genes encoded in the *V. cholerae* genome. We therefore refined the screen by pre-selecting for mutants that retained motility on soft agar, followed by a subsequent screen for loss of pellet formation in the motile subset of the mutant pool, as described above (**Fig 2A**). Interestingly, 19 of the 34 transposon insertions answering this screen mapped to the Vibrio Seventh Pandemic island (VSP-II) (**Figs 2B** and **S2A and S2B**), a horizontally acquired genomic region that is strongly associated with the El Tor biotype and the current (seventh) cholera pandemic. Transposons concentrated in a section of VSP-II that encodes a putative AraC-like transcriptional activator (VC0513, henceforth "VerA"), two ligand-sensing

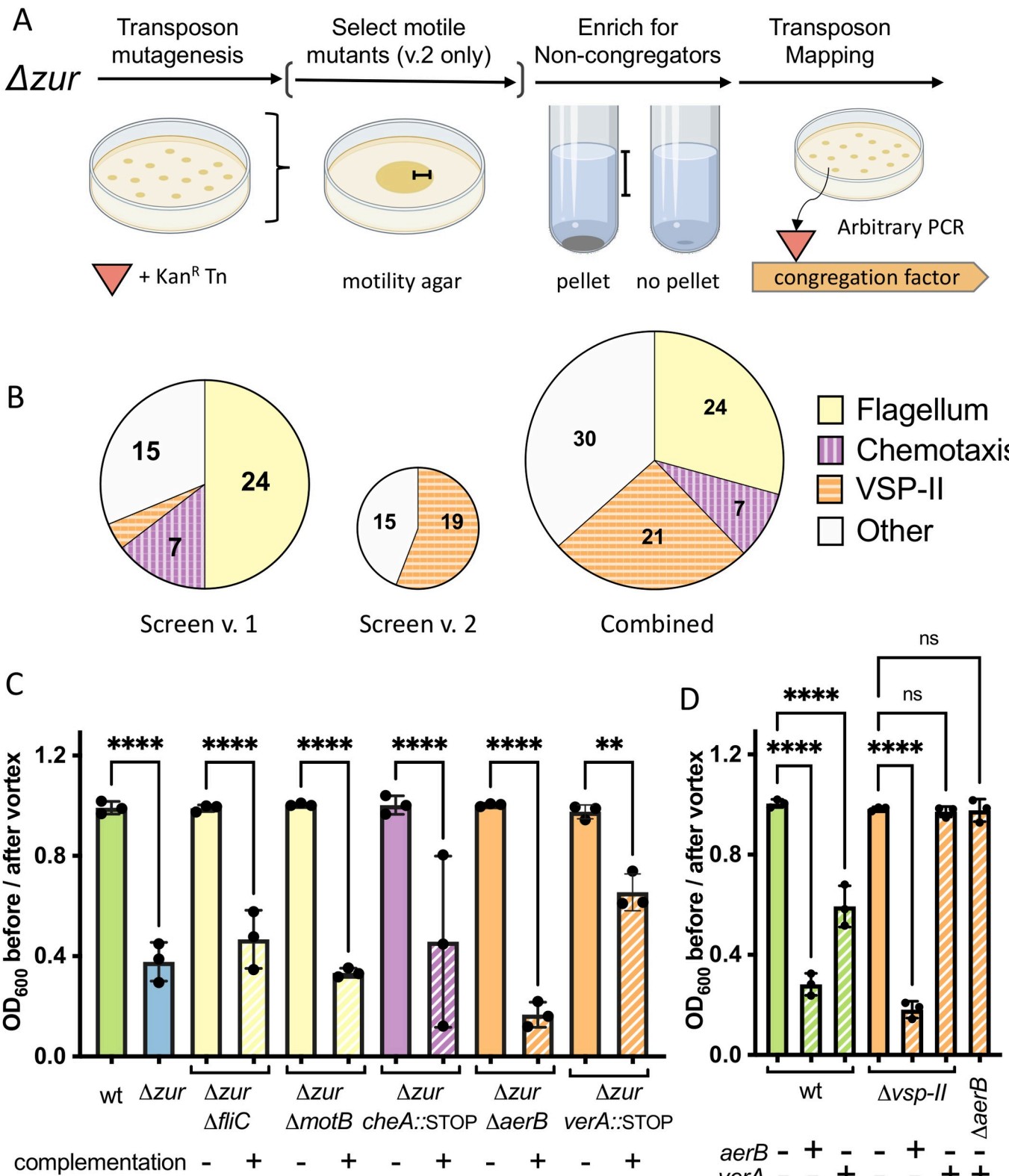

**Fig 2. *Δzur* congregation requires motility and components of VSP-II.** (**A**) *Δzur* was mutagenized with mariner transposons to generate a library of insertion mutants (see Methods for details). Non-congregating mutants within the library were enriched via repeated subculturing of the supernatant until no pellet formed. Brackets indicate harvested zones. Transposon insertions were mapped using arbitrary PCR. In a modified version of this screen (v.2), the transposon

library was pre-filtered to select for motile mutants on soft agar (0.3%). Schema created with BioRender.com. (**B**) Transposon insertions mapped to genes encoding flagellar components and regulators (24 hits, yellow), chemotaxis proteins (7 hits, purple/vertical lines), and VSP-II genes (21 hits, orange/horizontal lines). (**C**) Select motility (*fliC, motB*), chemotaxis (*cheA*), and VSP-II genes (*vc0512/aerB, vc0513/verA*) were mutated in a *Δzur* background (solid bars) and complemented back *in trans* (+) under an IPTG inducible promoter integrated within chromosomal *lacZ*. (Note: P$_{iptg}$-*verA* on a multicopy plasmid was used for complementing the *Δzur verA::STOP* mutant. These cultures were grown with kanamycin to ensure retention of either the empty or *verA*-expressing plasmid). Congregation in M9 glucose (0.2%) supplemented with inducer (IPTG, 100 μM) was quantified by measuring the optical density (at 600 nm) of the culture supernatant before and after a brief vortex. (**D**) Integrated, chromosomal copies of *aerB* or *verA* were overexpressed in wild-type and *Δvsp-II* (or a *ΔaerB*) backgrounds in M9. Ten and 200 μM of IPTG were used for *aerB* and *verA* induction, respectively. For all plots, the shown raw data points are biological replicates, error bars represent standard deviation, and asterisks denote statistical difference via Ordinary one-way ANOVA test (****, p < 0.0001; **, p < 0.01; n. s., not significant).

chemotaxis proteins (VC0512, formerly Aer-1 is henceforth referred to as "AerB", and VC0514), and a cyclic di-GMP phosphodiesterase (VC0515). Notably, the *verA/vc0513-vc0515* operon is preceded by a canonical Zur binding site and is thus a novel candidate for Zur-dependent regulation.

To validate the VSP-II genes' involvement in *Δzur* congregation, we inactivated each gene in a *Δzur* background through either clean deletion or through insertion of a premature stop codon. *ΔaerB* and *verA::STOP* mutations prevented *Δzur* from congregating, whereas respective complementation with *aerB* and *verA* restored the pellet (**Fig 2C**). We noted that although *verA* and *aerB* were both required for congregation, transposon hits were concentrated in *verA*. Since a *Δzur verA::STOP* mutant yields a significantly increased swarm diameter on soft agar (**S2C Fig**), we speculate that *verA* insertions were overrepresented in the motile subset of our transposon library. We additionally tested a deletion of the entire VSP-I island and mutations in all other open-reading frames on VSP-II (including *vc0514* and *vc0515*), none of which were required for *Δzur* congregation under the conditions tested (**S1H Fig**).

To determine if either *verA* or *aerB* are sufficient to generate congregates, we overexpressed each gene in a wild-type and a *Δvsp-II* background. Both *aerB and verA* overexpression caused the wild-type to congregate, but only the *aerB* chemoreceptor triggered congregation in a strain lacking other VSP-II genes (**Fig 2D**). These data suggest that AerB drives the observed pellet formation and raise the possibility that VerA functions as a transcriptional activator of *aerB*. Altogether, these two screens indicate that pellet formation in *Δzur* is driven by chemotactic flagellar movement, with assistance from a VSP-II encoded transcriptional activator (VerA) and chemoreceptor (AerB). These results were intriguing given that very little is known about the regulation or function of VSP-II encoded genes.

### Several VSP-II genes are significantly upregulated in a Δzur mutant

Prior inquiry into VSP-II function was made difficult by a lack of native gene expression under laboratory conditions; thus, we prioritized mapping the transcriptional networks embedded in this island. We predicted that the VSP-II genes of interest are expressed in *Δzur*. Consistent with this idea, the *verA* promoter region contains a highly conserved Zur-binding sequence approximately 200 bp upstream of the mapped transcription start site (determined by 5'-RACE in a *Δzur* mutant, **Figs 3A and S3**). Although the distance between the Zur box and transcriptional start site is greater than that observed for most *V. cholerae* Zur targets, equivalent or greater distances are noted for the Zur-regulated *ribA* (140–210 bp upstream of the ORF) and *zbp* (~380 bp upstream of ORF) in closely related *Vibrio spp.*, respectively [35]. To verify regulation by Zur, we measured *verA* promoter activity via a *lacZ* transcriptional fusion (P$_{verA}$-*lacZ*). LacZ encodes β-galactosidase (LacZ) and, when expressed, produces a colorimetric readout in the presence of a cleavable substrate (e.g., ONPG). As expected, transcription from the *verA* promoter in zinc-rich LB medium was robust in *Δzur* relative to wild-type or a *zur* complemented strain (**Fig 3B**). This data indicates that Zur negatively regulates *verA*

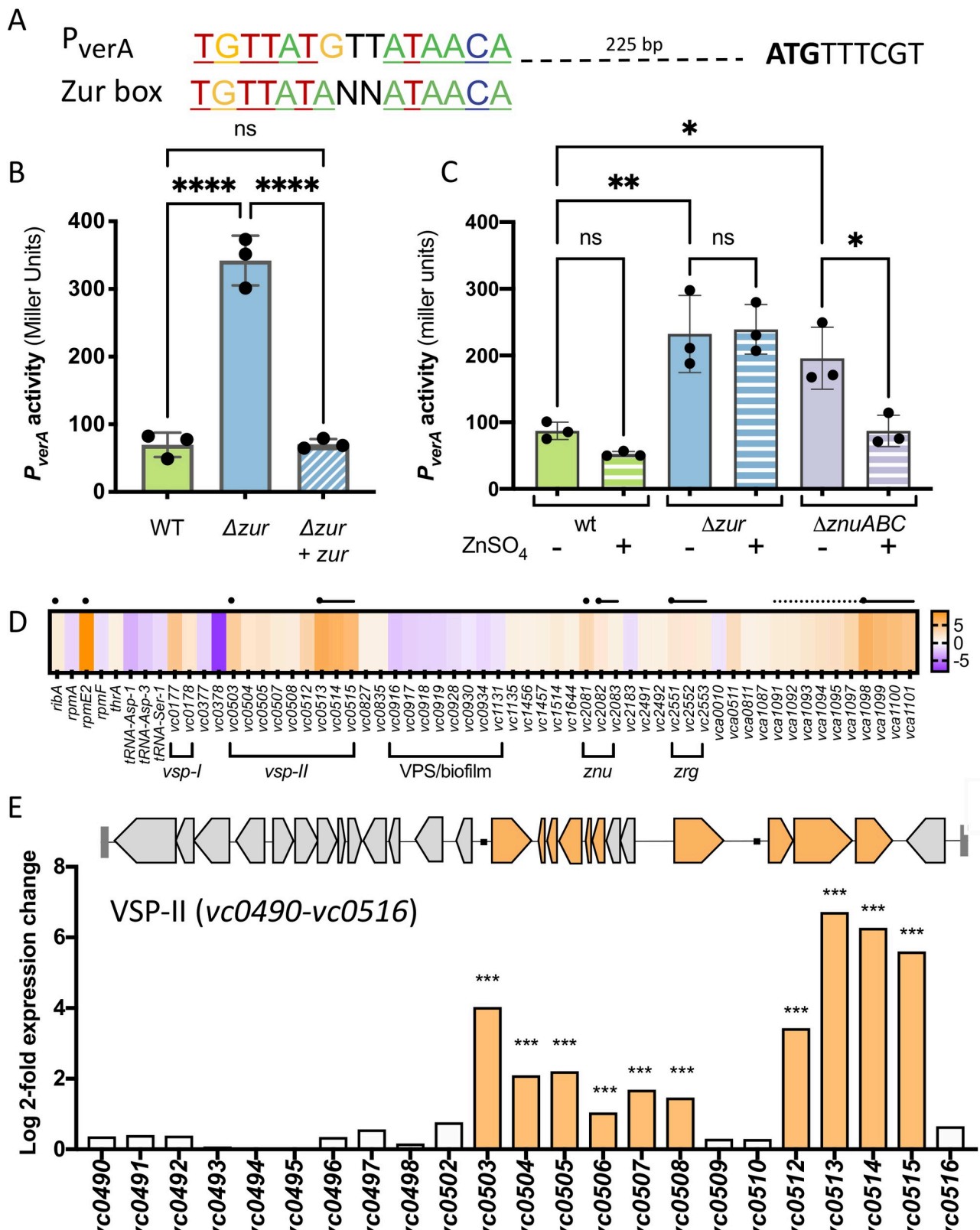

**Fig 3. Several VPS-II genes are upregulated in a Δ*zur* mutant. (A)** A predicted Zur-binding site (TGTTATGTTATAACA) located approximately 200 bp upstream of the *verA* open reading frame was aligned with the *Vibrionacae* Zur binding consensus sequence [35]. The predicted start codon (ATG)

is indicated in bold. (**B**) $P_{verA}$-*lacZ* transcriptional reporters were introduced into a wild-type and *Δzur* background, paired with either an empty vector or IPTG-inducible copy of *zur* (+ *zur*). Strains were grown overnight and diluted 1:100 into LB with kanamycin and inducer (IPTG, 200 μM). After 3 hours of growth at 37˚C (to mid/late exponential phase), promoter activity was quantified in Miller Units by measuring β-galactosidase (LacZ) activity against an ONPG chromogenic substrate (see Methods for more details). (**C**) Wild-type, *Δzur*, and *ΔznuABC* mutants carrying the $P_{verA}$-*lacZ* reporter were grown in M9 minimal medium in the presence (+) and absence (-) of exogenous zinc ($ZnSO_4$, 1 μM). After overnight growth (~16 h), promoter activity was measured in Miller units. For bar graphs, raw data points represent biological replicates, error bars represent standard deviation, and asterisks denote statistical difference via Ordinary one-way ANOVA (****, $p < 0.0001$; **, $p < 0.01$; *, $p < 0.05$; and n.s., not significant). (**D-E**) RNA was isolated from wild-type and *Δzur* cells at mid-log phase and prepared for RNA-seq (see Methods and Materials). Genes with significant differential expression in *Δzur* (log 2-fold change > 1, adjusted p-value <0.05) relative to wild-type N16961 are shown. (**D**) The heat map indicates increased (orange) or decreased (purple) expression relative to the wild-type strain. Black circles represent putative Zur binding sites and lines correspond to likely operons. (**E**) Log 2-fold expression changes for all VSP-II genes (*vc0490-vc0516*) are shown alongside a schematic of VSP-II open reading frames. Significantly upregulated genes are shown in orange.

transcription in a rich medium. We also tested $P_{verA}$-*lacZ* expression in M9 minimal medium in a wild-type, *Δzur*, and *ΔznuABC* background and noted that promoter activity corresponded to the conditions in **Fig 1D** that triggered congregation (**Fig 3C**). $P_{verA}$ activity was low in the wild-type background, suggesting that *V. cholerae* is not zinc-deficient in our M9 liquid culture. In contrast, $P_{verA}$ activity was robust in both *Δzur* and a mutant deficient in zinc uptake (*ΔznuABC*). Consistent with Zur's zinc sensing function, *Δzur* $P_{verA}$-*lacZ* strain retained high levels of β-galactosidase activity regardless of zinc availability, whereas $P_{verA}$-*lacZ* in *ΔznuABC* was repressible with exogenous zinc. These data, in conjunction with the highly conserved Zur binding site, suggest that the VerA-encoding *vc0513-vc0515* operon is a novel component of the Zur-regulated zinc starvation response in N16961.

Global transcriptomic studies of the Zur regulon have been conducted in a number of bacteria, but none thus far have been reported in the *Vibrio* genera [51–68]. We thus performed an RNA-seq experiment comparing transcript abundance in wild-type N16961 and *Δzur* to assess *V. cholerae*'s Zur regulon more comprehensively (including indirect effects). To ensure sufficient repression of Zur targets in the wild-type, cells were grown to mid-log phase in LB medium. Analyses identified 58 differentially expressed genes in *Δzur* (log 2-fold change >1, adjusted p-value <0.05) (**Figs 3D and 3E and S4 and S1 Table**). Seven promoters (situated in proximity to 23 of the 42 upregulated genes) contained an upstream canonical Zur-binding site. Among them were known or inferred (based on *E. coli*) Zur regulon components, including genes that encode zinc uptake systems (ZnuABC, ZrgACD), an alternative ribosomal protein (RpmE2), and a GTP cyclohydrolase (RibA). This transcriptomic analysis also uncovered what appears to be a bidirectional promoter with a Zur box: this locus encodes a strongly upregulated ABC-type transporter (*vca1098-vca1101*) in one direction and upregulated portions of the chemotaxis-3 (*che-3*) cluster (*vca1091-vca1095*, *vca1097*) in the other. Using a *lacZ* transcriptional reporter, we verified that the ABC-type transporter is indeed Zur-regulated (**S5 Fig**). Neither the transporter nor the *che-3* cluster (*vca1090-vca1097*), however, were required for *Δzur* congregation (**S1C Fig**). We observed a striking cluster of ten up-regulated genes on VSP-II (comprising 33% of the annotated open reading frames on VSP-II), including the previously characterized peptidoglycan hydrolase ShyB (encoded by *vc0503*) and the verA/*vc0513-vc0515* operon (**Fig 3D and 3E**). These results are consistent with both our transcriptional fusion data and the transposon screen.

The RNA-seq analysis also identified 36 differentially expressed genes that lacked canonical Zur binding sites (**Figs 3D and 3E and S4 and S1 Table**). Nineteen of these genes were significantly up-regulated in *Δzur*, including several genes on VSP-II (*vc0504-vc0508* and *vc0512*) and VSP-I (*vspR/vc0177*, *capV/vc0178*). Other upregulated transcripts in *Δzur* encode for cholera toxin (*ctxA/B*), the toxin co-regulated pilus biosynthesis proteins (*tcpT/H*), and a chitin binding protein (*gbpA*). Seventeen genes were significantly down-regulated in *Δzur*, many of which were related to vibrio polysaccharide (VPS) synthesis and biofilm formation [69]. Thus,

a *zur* deletion affects numerous genes indirectly, possibly through Zur-dependent secondary regulators (e.g. VC0515, cyclic di-GMP phosphodiesterase; VerA, AraC-like transcriptional regulator), via secondary responses to the influx of zinc that the *Δzur* mutant is expected to experience, or via Zur-dependent small RNA interference. We did conduct a perfunctory analysis of small RNAs and encourage interested research communities to utilize our data deposited in NCBI GEO (**GSE173966**) to pursue additional lines of inquiry.

### VerA is a Zur-regulated transcriptional activator of aerB

Our mutational analyses above raised the possibility that the putative methyl accepting chemotaxis protein (MCP) AerB is controlled by the transcriptional activator VerA. Given the importance of AraC-family regulators in governing *V. cholerae*'s host-associated behaviors (e.g., ToxT, intestinal colonization and virulence [70–73]; Tfos, chitin-induced natural competence [74–76]) we sought to characterize the full VerA regulon. We performed an RNA-seq experiment comparing transcript abundance in N16961 overexpressing *verA*, relative to an empty vector control. Surprisingly, only three other genes were significantly upregulated (log 2-fold change >1, adjusted p-value <0.05): *vc0512/aerB*, *vc0514*, and *vc0515* (**Fig 4A** and **S2 Table**). We validated these findings using *lacZ* transcriptional reporters. Plasmid-mediated *verA* over-expression was sufficient to induce $P_{verA}$-*lacZ* in rich LB medium, suggesting that this operon is autoregulated by VerA (**Fig 4B**). To remove the autoregulatory effect of native VerA from our analysis, we performed additional measurements in a parent strain lacking VSP-II (and thus native *verA*). These data suggest that loss of Zur binding may lead to only a small increase in *verA* transcription, which is further amplified by a VerA-dependent positive feedback loop.

Although the *aerB* promoter lacks a conserved Zur binding site, our transcriptomic data suggests that Zur-regulated VerA promotes *aerB* transcription. To verify this, we constructed a $P_{aerB}$-*lacZ* transcriptional reporter. Our initial attempt using a small (400 bp) promoter fragment did not yield detectable signal under inducing conditions (**S6 Fig**). 5'-RACE mapping of the transcription start indicated that *aerB* is part of a much longer transcript (extending >1 kb upstream of the start codon). Thus, we designed a new reporter construct to include this entire region. $P_{aerB}$ activity in standard LB medium fell below our threshold for detection, but we found that VerA overexpression was sufficient to activate the *aerB* promoter (**Fig 4C**). $P_{aerB}$ was also strongly induced in a *Δzur* strain background–consistent with our initial RNA-seq–but only if the strain also carried a native or *trans* copy of *verA*. These data indicate that *aerB* expression is dependent upon VerA-mediated activation. In summary, we found that VerA is a Zur-regulated, transcriptional activator that upregulates four genes (*aerB*, *verA/vc0513-vc0515*) (**Fig 4D**).

### AerB mediates energy taxis and pellet formation

VerA-mediated induction of AerB drives *V. cholerae* to congregate in minimal medium. AerB is predicted to encode an MCP that senses concentration gradients of a particular ligand (either an attractant or repellent) and relays that signal via Che proteins that alter flagellar rotation [77]. To determine if AerB indeed functions as a chemotaxis receptor, we first tested whether AerB interacts with the chemotaxis coupling protein, CheW. In a bacterial two hybrid assay, AerB and CheW were each fused with one domain of the adenylate cyclase (AC) protein (T18 or T25) and co-transformed into an *E. coli* strain. If the proteins of interest interact, the proximal AC domains will synthesize cAMP and induce *lacZ* expression via a cAMP-CAP promoter; thus, a positive protein interaction will yield blue colonies in the presence of X-gal. *E. coli* co-transformed with T18-AerB and CheW-T25 (or the reciprocal tags) yielded bright blue spots (**Fig 5A**). We additionally detected strong protein-protein interaction between

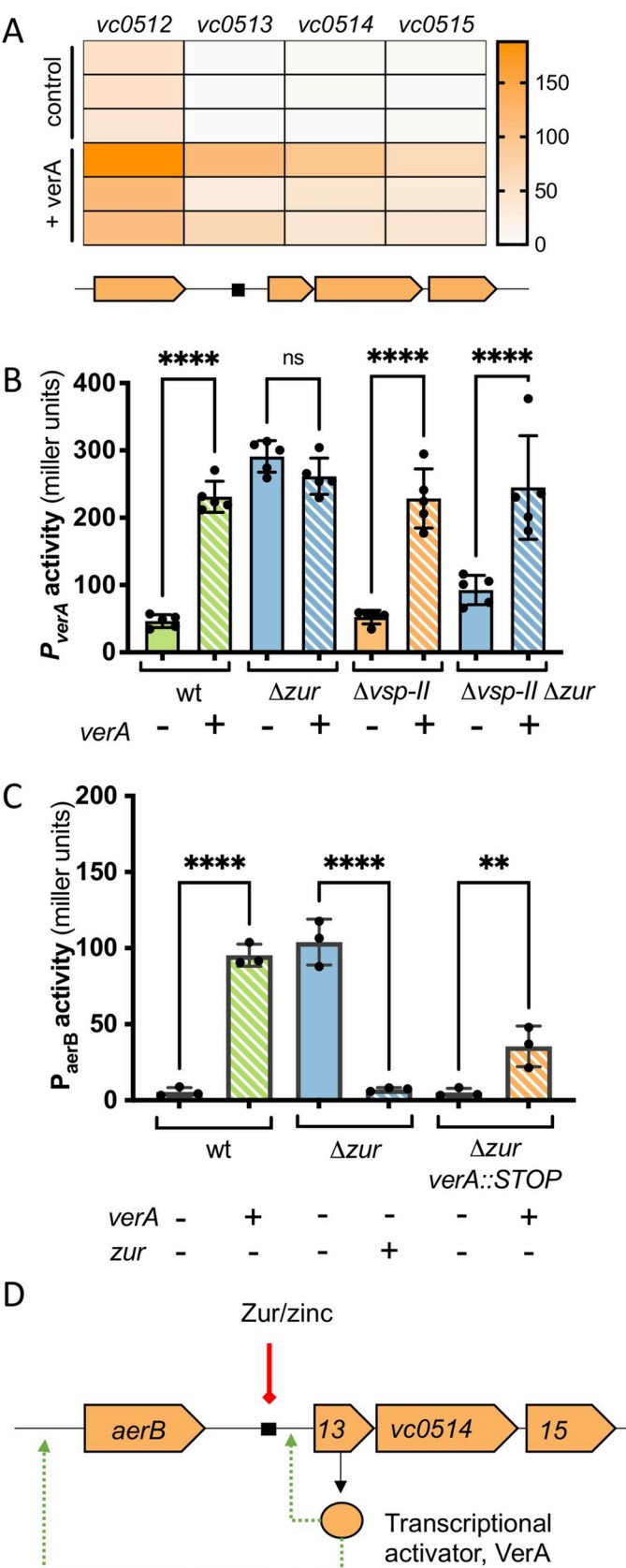

**Fig 4. VerA is an AraC-like transcriptional activator that positively regulates *aerB* and the *vc0513-vc0515* operon.**
(**A**) Overnight cultures of wild-type N16961 carrying either an IPTG-inducible copy of *verA* (+ *verA*) or empty vector (control) were diluted 1:100 in fresh LB containing kanamycin and IPTG (1 mM). RNA was isolated from cells at mid-log phase and prepared for RNA-seq (see Methods and Materials). Heat map shows normalized expression values for differentially expressed genes across three biological replicates. (**B-C**) Overnight cultures of strains carrying *lacZ* transcriptional reporters were diluted 1:100 in LB and grown for 3 h at 37˚C. Kanamycin and inducer IPTG (500 μM) were included in the growth medium for retention and *trans* expression (+) of *verA* or *zur* from an IPTG-inducible promoter. Promoter activity (in Miller Units) was measured via β-galactosidase assays (See Methods and Materials). (**B**) $P_{verA}$-*lacZ* activity was measured in wild-type, Δ*zur*, Δ*vsp-II*, and Δ*zur* Δ*vsp-II* strains carrying a plasmid-borne, IPTG-inducible copy of *verA* (+, striped bars) or an empty vector control (-, solid bars). (**C**) Activity from a $P_{aerB}$-*lacZ* reporter was measured in wild-type, Δ*zur*, and Δ*zur verA*::STOP backgrounds harboring a plasmid-borne, IPTG-inducible copy of *verA* or *zur* (+, striped bars) or an empty vector control (-, solid bars). (**D**) Proposed model for Zur repression of the *verA* promoter (solid line, red) via a conserved Zur binding site (black box) and subsequent VerA-dependent activation (dashed arrow, green) of the *aerB* and *verA* promoters.

T18-AerB and T25-AerB, indicating that our chemoreceptor can dimerize (or oligomerize) like other MCPs [78]. To confirm that AerB's MCP signaling domain is required for congregation, we next mutated a glycine residue within the highly conserved C-terminal hairpin loop (R-A-**G**-E/D-X-G) [79] of AerB (**S7 Fig**), which is required for *in vitro* signal generation in other MCPs [80]. A Δ*vsp-II* strain expressing this AerB[G385C] mutant was unable to congregate (**Fig 5B**), consistent with MCP function. Together, these data suggest that AerB indeed functions as a chemotaxis receptor.

Intriguingly, the chemical ligands for AerB and the vast majority of *V. cholerae*'s 46 encoded MCPs are yet to be determined [81]. The AerB N-terminus harbors a PAS domain [82], a protein family that typically senses light, oxygen, redox stress, or electron acceptors [83]. We hypothesized that this PAS-containing chemoreceptor mediates energy taxis, perhaps along the oxygen gradient in our vertical culture tubes. We first tested whether oxygen was required for Δ*zur* to congregate. Wild-type and Δ*zur* were cultured in both aerobic and anoxic tubes in M9 minimal medium with the terminal electron acceptor fumarate to enable anaerobic glucose respiration. In contrast with the aerobic cultures, Δ*zur* did not congregate under anoxic conditions (**Fig 5C**). A similar result was observed under glucose-fermenting conditions (i.e., when fumarate was omitted from the medium) (**S8 Fig**). These data indicate that the chemotaxis-driven congregation in Δ*zur* is oxygen-dependent.

AerB shares 31% amino acid identity with *V. cholerae*'s primary aerotaxis receptor Aer-2 (renamed here to AerA, as numbers in bacterial gene names can be confused with mutant alleles) (**S7A Fig**), which exhibits a positive response to oxygen [84]. Other homologs include *E. coli*'s Aer[EC] (B0372, 31% identity), which positively responds to oxygen via sensing the electron acceptor FAD [83,85,86]. Alignment of AerB with additional orthologs from *Azospirillum brasilense* [87] and *Shewanella oneidensis* [88] revealed conservation of a critical FAD-binding tryptophan residue, among others (**S7B Fig**). The corresponding amino acids were mutated in *aerB* and each mutant was expressed in a Δ*vsp-II* background to determine whether they still promoted congregation. Strains expressing the W74F mutant failed to congregate, suggesting that this highly conserved FAD-binding residue is essential for function (**Fig 5B**). Two additional AerB mutations (R61A and H62A), corresponding to *E. coli* FAD-binding residues, were also unable to congregate. The requirement for oxygen and these putative FAD-binding residues suggests that AerB may bind FAD or a similar ligand to facilitate energy taxis.

Since *aerB*-expressing cells congregated at the bottom of the culture tube, we hypothesized that AerB mediates a negative response to oxygen, in contrast to the positive response mediated by AerA and Aer[EC]. To further interrogate energy taxis, we examined the swarming dynamics of *V. cholerae* in an established aerotaxis assay, which uses soft agar with carbon sources that vary in their ability to accentuate aerotaxis behavior [84,85]. Succinate, for

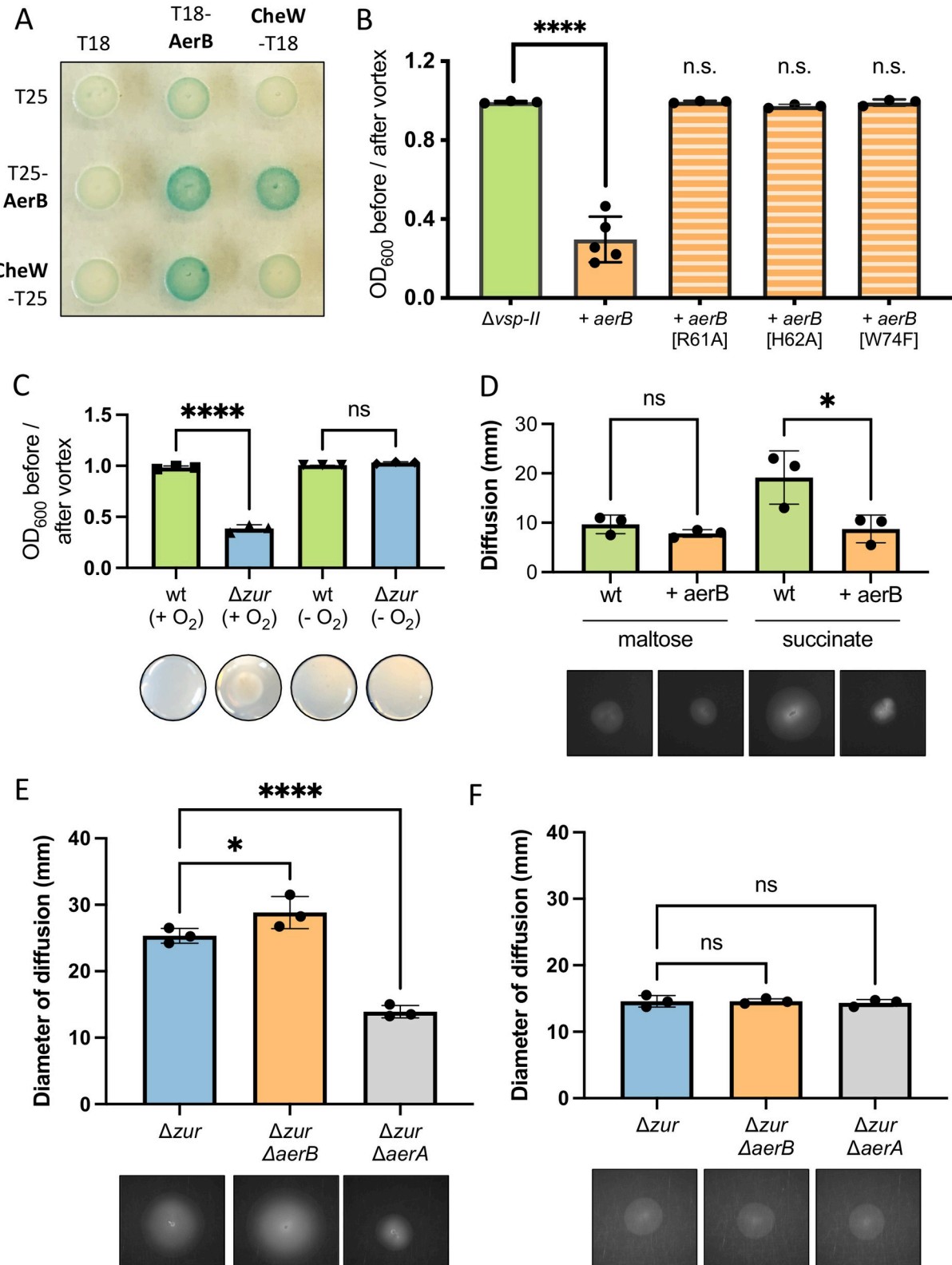

**Fig 5. AerB encodes a methyl-accepting chemotaxis protein involved in energy taxis.** (**A**) In a bacterial two-hybrid assay, *E. coli* BTH101 was co-transformed with vectors carrying one domain of adenylate cyclase (T18 or T25) or an adenylate cyclase fusion with a protein of interest: CheW-T(18/25) or T(18/25)-AerB. Co-transformants were spotted onto an LB agar containing kanamycin and ampicillin (for

selection), X-gal (for blue-white detection), and inducer (IPTG, 500 μM). Plates were incubated overnight at 30°C and for an additional day at room temperature. Blue color signifies positive protein-protein interactions. (**B**) *Δvsp-II* strains carrying an integrated, IPTG-inducible copy of either *aerB* or *aerB* point mutants (G385C, R61A, H62A, or W74F) were grown shaking overnight in M9 minimal medium supplemented with glucose (0.2%) and inducer (IPTG, 10 μM) at 30°C. Congregation was quantified by measuring the optical density (at 600 nm) of the culture supernatant before and after a brief vortex. (**C**) Wild-type and *Δzur* were grown overnight in 5 mL M9 minimal medium plus glucose (0.5%) and a terminal electron acceptor (fumarate, 50 mM). Cultures were grown aerobically (+ $O_2$) or anoxically (- $O_2$) shaking overnight at 30°C. Congregation was quantified as described above and representative images of the bottom of each culture tube are shown. (**D**) Wild-type and a strain carrying an inducible chromosomal copy of *aerA* (+ *aerA*) were grown overnight in LB medium and washed thrice in M9 minimal medium lacking a carbon source. A sterile toothpick was used to inoculate cells into M9 soft agar (0.3%) containing inducer (IPTG, 10 μM) and either succinate (30 mM) or maltose (0.1 mM) as a carbon source. The diameter of diffusion (mm) was measured following a 48-hr incubation at 30°C and representative diffusion patterns are shown for each strain. (**E-F**) Motility of *Δzur*, *Δzur ΔaerB*, and *Δzur ΔaerA* were measured as described above in (**E**) succinate and (**F**) maltose plates. Representative swarm diameters are shown. For all bar graphs, raw data points represent biological replicates, error bars represent standard deviation, and asterisks denote statistical difference via Ordinary one-way ANOVA test (****, $p < 0.0001$; *, $p < 0.05$; n.s., not significant).

example, can only be catabolized via respiration, which consumes oxygen, thereby generating an $O_2$ gradient that increases with distance from the inoculation site. Since no other classical attractants/repellents are present, motility on succinate plates reveals aerotaxis as the primary taxis behavior [85]. In contrast, maltose agar provides other cues for chemotaxis (including chemotaxis towards maltose itself), obscuring an aerotactic response. The diameter of diffusion in succinate and maltose (at 30°C) was measured two days post-inoculation. Induction of *aerB* in a wild-type background significantly reduced the swarming diameter in succinate (**Fig 5D**). In contrast, *aerB* induction did not affect the swarming diameter in maltose plates. These data suggest that AerB promotes a negative chemotactic response to (or in the presence of) oxygen gradients.

Additional assays were performed in a *Δzur* background to ensure robust expression of *aerB* from the native promoter. *aerA* mutants were included as a control. Compared to *Δzur*, the diameter of *Δzur ΔaerB* migration on succinate was subtly, but significantly, increased (**Figs 5E** and **S9A and S9B**). This is consistent with AerB promoting a negative response to oxygen. Conversely, *Δzur ΔaerA* showed a significant decrease in swarming ability, consistent with previous reports that showed AerA promoting a positive response to oxygen [84]. There were no significant differences between the swarm diameter of *Δzur* and the *aer* mutants on maltose plates (**Figs 5F** and **S9C and S9D**). These assays were additionally performed in a wild-type background and *aerB* showed no effect on swarming behavior (**S9 Fig**), consistent with lack of *aerB* transcriptional expression in wild-type background. These results are consistent with AerB functioning as an energy taxis receptor that is either dependent on, or responsive, to oxygen.

## Ectopic AerB induces congregation in El Tor strains with atypical VSP-II islands

Although the VSP-II island is strongly correlated with the 7[th] pandemic strain, variants of this island have been detected in other El Tor isolates (**Fig 6A**). The Zur-regulated VSP-II genes characterized in this study appear to be in a hotspot for island variation: C6706 (Peru, 1991) lacks *vc0511-vc0515* while the Haiti strain (2010) lacks *vc0495-vc0512*. We predicted that only El Tor *Δzur* strains with prototypical islands (harboring *aerB*/*vc0512* and *verA*/*vc0513*) will congregate in minimal medium. As expected, the prototypical VSP-II strains (N16961 and E7946) lacking *zur* congregated in overnight culture (**Fig 6B**). This suggests that *Δzur* congregation is likely not due to strain-specific variations outside of VSP-II. In contrast, neither the C6706 nor the Haiti mutant congregated in minimal medium, presumably due to the absence of *aerB*. To test whether we could promote congregation in these VSP-II variants, we expressed a chromosomally integrated, inducible copy of *aerB* in C6706 and Haiti. Similar to the N16961

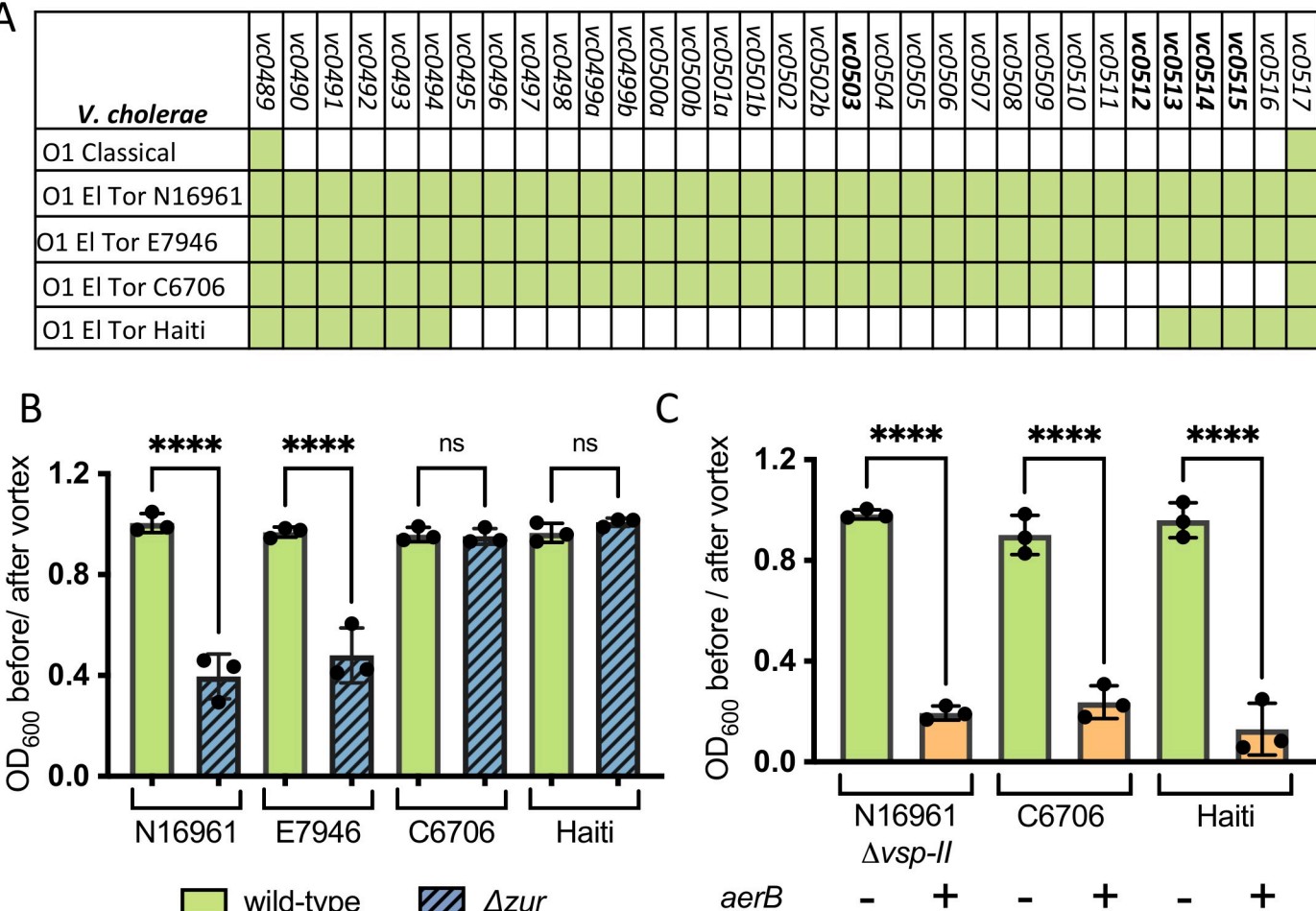

**Fig 6. Ectopic AerB expression causes atypical El Tor strains to congregate.** (A) Presence (green) or absence (white) of VSP-II open reading frames in *V. cholerae* strains with prototypical (N16961, E7946) and variant (Haiti, C6706) VSP-II islands are shown. (B) Wild-type or Δ*zur* varieties of *V. cholerae* (N16961, E7946, C6706, Haiti) were grown overnight (~15 hr) shaking at 30˚C in M9 minimal medium with glucose (0.2%). Congregation was quantified by measuring the optical density (at 600 nm) of the culture supernatant before and after a brief vortex. (C) *V. cholerae* strains with (+) or without (-) an IPTG-inducible, chromosomal copy of *aerB* were grown as described in **B** but with the addition of IPTG (10 uM). For all bar graphs, raw data points represent biological replicates, error bars represent standard deviation, and asterisks denote statistical difference via Ordinary one-way ANOVA (****, $p < 0.0001$; n.s., not significant).

Δ*vsp-II* control, C6706 and Haiti expressing *aerB* congregated in minimal medium (**Fig 6C**). These data suggest that AerB's interaction partners are conserved in other *V. cholerae* El Tor isolates.

### A model for oxygen-dependent V. cholerae congregation in zinc starved environments

In summary, we propose the following model for Δ*zur* congregation in M9 minimal medium (**Fig 7**). In zinc rich conditions, Zur acts as a repressor of the VerA-encoding *vc0513-vc0515* operon on VSP-II. In the absence of Zur or in zinc starvation, the VerA transcriptional activator induces expression of its own operon and the nearby *aerB*. AerB serves as a receptor for oxygen-dependent energy taxis and relays changes in signal concentration to the core chemotaxis machinery and the flagellum. This results in cells congregating at the bottom of the culture tube in an oxygen-dependent manner.

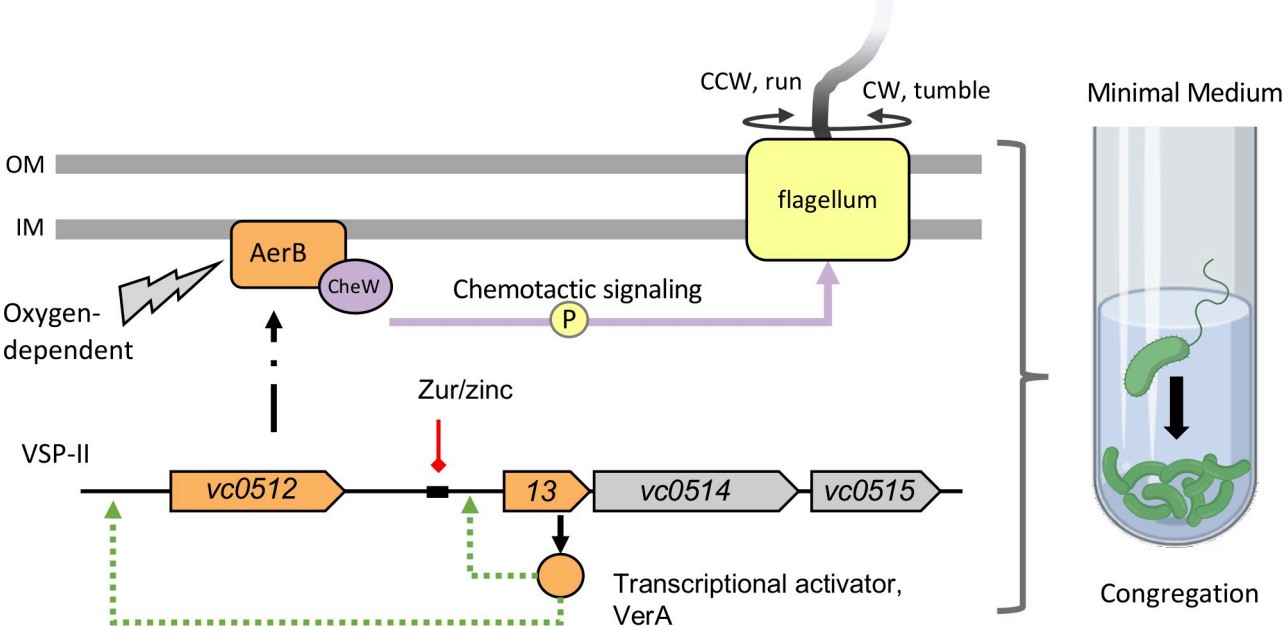

**Fig 7. Proposed model of Zur-regulation of VSP-II encoded genes and its effect on chemotaxis.** Zur forms a complex with divalent zinc ions and binds with high affinity to specific DNA sequences (black rectangle), repressing transcription of downstream genes (red line). In the absence of *zur* or during zinc starvation, VSP-II genes, including the *vc0513-vc0515* operon are derepressed. The vc0513-encoded transcriptional activator VerA induces transcription (green arrow) of *aerB*, which encodes a chemotaxis receptor. AerB interacts with the chemotaxis coupling protein CheW (purple) and mediates a signal relay that alters flagellar (yellow) rotation and cell reorientation. AerB induction in minimal medium causes *V. cholerae* to congregate in an oxygen-dependent manner away from the air-liquid interface. Model created with BioRender.com.

## Discussion

The mysterious Vibrio Seventh Pandemic Island (VSP-II) present in the El Tor biotype has largely evaded characterization due to lack of knowledge of stimuli that favor its induction. We report that Zur, the transcriptional repressor of the zinc starvation response, is a direct and indirect regulator of other VSP-II genes. Novel Zur targets reported here include the *vc0513-vc0515* operon, which encodes the VerA transcriptional activator that increases expression of VSP-II chemotaxis and motility-related genes. One of these secondary targets, AerB, encodes a chemoreceptor involved in energy taxis.

### The role of zinc availability in VSP-II induction

It has long been suspected that the VSP islands function as either pathogenicity or environmental persistence islands. Recent work interestingly suggests that VSP-I may function as a phage defense system [89,90]. In contrast, we and others have not yet identified a set of growth conditions under which VSP-II confers a fitness benefit [23,25,91,92] (**S10 Fig**). Given the robust expression of VSP-II loci in the absence of Zur, we propose two contexts where *V. cholerae* may encounter zinc starvation and express these island-encoded genes: within the human host, and/or on (chitinous) biotic surfaces in aquatic reservoirs.

The human host is a well-studied example of a metal-limited environment. Vertebrate hosts sequester desirable metal cofactors (e.g. zinc) in order to restrict the growth of potentially harmful bacteria (i.e. nutritional immunity, [29–32]). Pathogens lacking zinc acquisition systems often exhibit colonization defects *in vivo* [34,93–100], potentially because they are unable to compete against the microbiota for precious metal cofactors [101]. Induction of zinc starvation genes in pathogenic *V. cholerae* appears to be dependent upon the *in vivo* model used; for

example, the primary zinc importer and the *vc0513-vc0515* operon are upregulated in a mouse but not in a rabbit model (relative to LB) [28]. In terms of fitness, *in vivo* infection assays showed that loss of *V. cholerae*'s zinc importers led to modest colonization defects in both mouse [34] and rabbit infection models [102]; however, the latter Tn-Seq analysis performed in a rabbit model did not detect any significant fitness defects in VSP-II mutants [102]. In summary, although VSP-II genes may be expressed in certain animal models, it remains unclear if the island confers any advantage during host infection.

More generally, *V. cholerae* experiences metal starvation within thick bacterial communities and thus metal transporters and regulators contribute optimal *V. cholerae* biofilm formation [103]. Zur-regulated genes (including *vc0503* and *vc0513-vc0515*) are reportedly induced by exposure to chitin oligomers [33], raising the possibility that *V. cholerae* is zinc-limited while colonizing copepods or crustaceans in the environment. It is thus plausible that the Zur-regulated VSP-II genes may be involved in *V. cholerae*'s chemotactic movements within the aquatic reservoir.

## VSP-II-encoded genes facilitate chemotactic responses

Connections between zinc homeostasis and altered motility patterns have been reported in other bacteria, but these phenotypes appear to be indirect consequences of zinc availability rather than Zur repression of secondary transcriptional regulators [97,104–106]. The VerA-regulated chemoreceptor, AerB, generates congregation in liquid culture and appears to mediate energy taxis. This is in apparent contradiction with a report that did not find a role for AerB (referred to as *Aer-1*) in aerotaxis [84]; however, this may be explained by a lack of native *aerB* expression under their experimental conditions. *aerA* expression, on the other hand, does not appear to be regulated by Zur and native levels are sufficient to alter motility of the wild-type on soft agar. The balanced action of aerotactic responses conferred by AerA and AerB may function analogously to the Aer and Tsr receptors [107], which enable *E. coli* to navigate to an optimum oxygen concentration. However, we do not exclude the possibility that the AerB chemotactic response is more complex than the model proposed here.

Although the role of chemotaxis in autoaggregation has not been previously reported in *V. cholerae*, this phenomenon has been characterized in several distantly related bacteria [108]. *A. brasilense*, for example, aggregates in response to oxygen/redox stress via a PAS-containing chemoreceptor homologous to AerB (33% amino acid identity, **S7A Fig**) [87,109,110]. As a second example, *Shewanella oneidensis* "congregates" around insoluble electron acceptors [111] via an MCP with a PAS domain (SO_1385, 39% amino acid identity to AerB), an MCP with a $Ca^{2+}$-sensing Cache domain (SO_2240, 39% amino acid identity to VC0514), and a protein involved in extracellular electron transport (CymA, SO_4591) [88]. This study, along with a correlogy analysis of VSP genes [90], suggests that the AerB/VC0512 and VC0514 MCPs may be functionally linked. Future work will investigate the relationship between these VerA-regulated chemotaxis receptors.

The energy taxis system described here may enable *V. cholerae* to avoid redox stress in low-zinc environments, since deletion of zinc importer systems is associated with heightened redox susceptibility in *E. coli* [98]. We speculate that AerB may allow *V. cholerae* to colonize other niches within the host (e.g., anaerobic parts of the gut), similar to a redox-repellent chemotaxis system in *Helicobacter pylori* that enables gland colonization *in vivo* [112,113]. Alternatively, this chemotaxis system may allow *V. cholerae* to exploit different niches within the aquatic reservoir (e.g., anoxic sediments with chitin detritus); however, each of these biologically relevant conditions are difficult to recapitulate *in vitro*.

Chemotaxis enhances virulence in a number of enteric pathogens, but this does not seem to generally hold true for *V. cholerae* [114]. In an infection model, non-chemotactic (counter-

clockwise biased) mutants outcompeted wild-type *V. cholerae* and aberrantly colonized parts of the upper small intestine [115], suggesting that chemotaxis is dispensable and possibly deleterious for host pathogenesis. *V. cholerae* appears to broadly downregulate chemotaxis genes in a mouse infection model [28] and in stool shed from human patients [116]. Intriguingly, this decrease may be mediated in part by VSP-I; the island-encoded DncV synthesizes a cyclic AMP-GMP signaling molecule that decreases expression of chemotaxis genes and enhances virulence [24]. Specific chemoreceptors, however, are upregulated within a host and/or enhance virulence (see [117] for a review). Given the conflicting roles for chemotaxis within a host and the lack of evidence for VSP-II's role during infection, we alternatively suggest that VSP-II encoded chemotaxis genes may serve a purpose in an aquatic environment with oxygen and nutrient gradients.

## Stress and starvation responses can be co-opted by acquired genetic elements

We report that targets of the Zur-regulated VerA appear to be restricted to VSP-II, at least under the conditions tested. This restriction is logical given that transcriptional activators require specific DNA-binding sequences, and these may not be present in the native chromosome of a horizontal transfer recipient. Zur control of secondary regulators, including those that impact gene expression more broadly via signaling molecules (i.e., cyclic di-GMP phosphodiesterases like VC0515) may function to expand the complexity and tunability of the Zur-regulon in response to zinc availability and compounding environmental signals.

Horizontal acquisition of genomic islands can help bacteria (and pathogens) evolve in specific niches. VSP-II retains the ability to excise to a circular intermediate in N16961, indicating the potential for future horizontal transfer events [26]. We observed that 33% of the ORFs on the prototypical VSP-II island are expressed in the absence of Zur. Intriguingly, genomic island "desilencing" in response to zinc starvation has been reported in diverse bacterium, including *Mycobacterium avium* ssp. Paratuberculosis [67] and *Cupriavidus metallidurans* [55]. We note in this study that other El Tor isolates lack some Zur-regulated components of VSP-II. Given that the emergence of VSP-II containing 7th pandemic strains is recent on an evolutionary timescale, the contents of these islands may still be undergoing selection.

**In summary**, investigation of our Zur-associated congregation phenotype enabled identification of novel components of the zinc starvation response present on the El Tor Vibrio Seventh Pandemic Island -II (VSP-II). Further characterization of these island-encoded genes may aid in establishing VSP-II's role as either a pathogenicity or environmental persistence island.

## Methods

### Bacterial growth conditions

Bacterial strains were grown by shaking (200 rpm) in 5 mL of LB medium at 30°C (for *V. cholerae* and *E. coli BTH101*) or 37°C (for other *E. coli*) in borosilicate glass tubes, unless otherwise specified. M9 minimal medium with glucose (0.2%) was prepared with ultrapure Mili-Q water to minimize metal contamination. Antibiotics, where appropriate, were used at the following concentrations: streptomycin, 200 μg ml$^{-1}$; ampicillin, 100 μg ml$^{-1}$, and kanamycin, 50 μg ml$^{-1}$. IPTG was added to induce P$_{iptg}$ promoters at indicated concentrations.

### Plasmid and strain construction

For all cloning procedures, N16961 gDNA was amplified via Q5 DNA polymerase (NEB) with the oligos summarized in **S3 Table**. Fragments were Gibson assembled [118] into restriction-

digested plasmids. For gene deletions, 700 bp flanking regions were assembled into XbaI-digested pCVD442 (Amp^R). For complementation experiments, genes of interest were amplified with a strong ribosome binding site and assembled into SmaI-digested pHLmob (kan^R) or pTD101 (Amp^R) downstream of an IPTG-inducible promoter. *lacZ* transcriptional reporters were built by amplifying the desired promoter region and assembling into NheI-digested pAM325 (Kan^R). The resulting promoter-*lacZ* fusions were amplified for assembly into StuI-digested pJL1 (Amp^R). Cloning for bacterial two-hybrid assays are described in a separate section below. All assemblies were initially transformed into DH5alpha λ*pir* and subsequently into an *E. coli* donor strain (MFD λ*pir* or SM10 λ*pir*).

For conjugations into *V. cholerae*, stationary phase recipients and MFD λ*pir* donor strains were washed of antibiotics, mixed in equal ratios, and spotted onto an LB DAP plate. After a 4 hr incubation at 37°C, cells were streaked onto an LB plate with ampicillin or kanamycin to select for transconjugants. Conjugations using SM10 λ*pir* donors were performed in the absence of DAP and with the addition of streptomycin to selective plates. pHLmob transconjugants were purified on an additional kanamycin LB agar plate. Integration vectors (pCVD442, pTD101, and pJL1) were cured through two rounds of purification on salt-free sucrose (10%) agar. Gene deletions or STOP codon replacements introduced by pCVD442 were verified by PCR using the oligos indicated in **S3 Table**. Successful integration of *lacZ* targeting vectors (pTD101, pJL1) were identified by blue-white screening on plates containing 5-Bromo-4-chloro-3-indolyl-β-d-galactopyranoside (X-Gal, 40 μg ml^−1). pJL-1 vectors were additionally checked via PCR. All strains used in this study are summarized in **S3 Table**. *V. cholerae* strains were derived from N16961, unless otherwise indicated as E7946 [119], C6706 (not strep^R) [14], or Haiti [19]. The N16961 accession numbers for genes referenced in this study are as follows: *zur/vc0378*, *znuABC/vc2081-vc2083*, *zrgABC/vc2551-2553*, *ribA/vc1263*, *rpmE2/vc0878*, *rpmJ2/vc0879*, *shyB/vc0503*, *aerB/vc0512*, *verA/vc0513*, *fliC/vc2199*, *motB/vc0893*, *cheA-2/vc2063*, *cheW-1/vc2059*, *cheZ/vc2064*, *cheY/vc2065*, *vpsL/vc0934*, *csqS/vca0522*, *csqA/vc0523*, *tdh/vca0885*, *luxS/vc0557*, *luxQ/vca0736*, and *aerA/vca0658*.

## Site-directed mutagenesis

Site-directed mutagenesis was performed using NEB kit #E0554S according to manufacturer instructions. A pTD101 plasmid carrying *aerB* was used as the template for Q5 amplification with the following mutagenic primer pairs: R61A, SM-1294/1295; and H62A, SM-1296/1297 (**S3 Table**). Products were purified and treated with kinase, ligase, and Dpn1 at 37°C for 30 minutes. This reaction mixture was transformed into DH5α λpir. Mutations were confirmed via Sanger Sequencing. *aerB* fragments containing W74F (SM-1306) and G395C (SM-1307) were chemically synthesized by Integrated DNA Technologies (IDT) and assembled into pTD101; sequences are listed in **S3 Table**.

## Congregation assays

Bacterial congregation was quantified by measuring absorbance (OD600) in a spectrophotometer of the culture supernatant before and after a brief vortex (5 seconds). Congregation score represents the ratio of before and after pellet disruption; a ratio closer to one indicates that the culture is homogenous, a ratio closer to zero indicates that the cells are concentrated at the bottom of the culture tube.

## Transposon mutagenesis screen & Arbitrary PCR

*V. cholerae* N16961 Δ*zur* was mutagenized with Himar1 mariner transposons via an SM10 λ*pir* donor strain carrying pSC189 [120]. Four independent Δ*zur* transposon libraries were

generated, as previously described [27]. Each library was separately harvested from the plate using sterile rubber scrapers, vortexed into LB, and preserved in glycerol at -80˚C. Individual culture tubes containing 5 mL of M9 minimal medium with glucose (0.2%) and kanamycin were inoculated with transposon libraries. Overnight cultures were back-diluted 1000-fold into fresh medium and incubated overnight; this process was repeated until no visible pellet had formed. Isolated colonies were tested to verify that they did not generate a pellet. The second screen was performed identically to the first, except that cultures were first inoculated into a M9 motility plate (0.3% agar) and allowed to migrate for 48 hours at 30˚C degrees. Scrapings from the outer zone (collected with a 1 mL pipette tip) were inoculated a culture tube containing M9 minimal medium. For both screens, the transposon insertion site for each isogenic colony was identified by arbitrary PCR [46]. As described elsewhere, this technique amplifies the chromosomal DNA adjacent to the mariner transposon. Amplicons were Sanger sequenced at the Cornell Institute of Biotechnology, and regions of high-quality were aligned to the N16961 reference genome using BLAST [121].

## RNA-seq and analysis

Overnight cultures of wild-type N16961 and the *Δzur* mutant were diluted 1:100 into LB and grown shaking at 37˚C until cells reached mid-log phase (optical density at 600 nm [$OD_{600}$], 0.5). RNA was extracted using mirVana miRNA Isolation Kit (Invitrogen, AM1560). Genomic DNA contamination was removed through two DNAfree (Ambion) treatments each followed by glass fiber column purification. Library preparations, Illumina sequencing, and data analysis were performed by GENEWIZ (South Plainfield, NJ). Differentially expressed genes were those with log 2-fold change >1 and an adjusted p-value <0.05. Raw and analyzed data files were deposited in the NCBI GEO database (**GSE173966**).

For VC0513 overexpression, wild-type N16961 carrying either pHLmob or pHLmob($P_{iptg}$-*vc0513*) was sub-cultured into LB kanamycin IPTG (500 µM) and grown at 37˚C for 3 hour (~ mid-log phase). Total RNA isolations and DNase treatments were performed as described above. Library preparations, Illumina sequencing, and data analysis (using DESeq2 [122]) were performed by the Cornell Transcriptional Expression Facility. Differentially expressed genes were those with log 2-fold change >1 and an adjusted p-value <0.05. Raw and analyzed data files were deposited in the NCBI GEO database (**GSE174028**).

## 5'-Rapid Amplification of cDNA Ends (5'-RACE)

Transcription start sites were identified with 5′-RACE. To obtain *vc0512*, *vc0513*, and *vca1098* transcripts, the *Δzur* mutant was grown in LB at 37˚C until cells reached mid-log phase (optical density at 600 nm [$OD_{600}$], 0.5). RNA extractions and DNAse treatments were performed as described for RNAseq. PCR was performed to check for genomic DNA contamination; no amplicons were detected within 34 cycles. Reverse transcription was performed with the Template Switching Reverse Transcriptase enzyme mix (NEB #M0466) according to manufacturer protocols using gene specific primers (*vc0512*, SM-1133; *vc0513*, SM-1131; *vca1098*, SM-1129) and the Template Switching Oligo (TSO). PCR Amplification of 5'-transcripts was performed with diluted cDNA, Q5 Hot Start High-Fidelity Master Mix (NEB #M0494), TSO-specific primer, and gene-specific primers (vc0512, SM-1134; vc0513, SM-1132, *vca1098*, SM-1130). Products were sanger sequenced using the following primers: SM-1134, SM-1156, and SM1157 for *vc0512*, SM-1132 for *vc0513*, and SM-1130 for *vca1098*. Primer sequences are listed in **S3 Table**.

## β-galactosidase activity measurements

*V. cholerae* strains carrying promoter-*lacZ* fusions were grown overnight in LB at 30˚C, with kanamycin for plasmid (pHLmob) maintenance. Strains were diluted 1:100 into LB containing kanamycin and IPTG (1 mM) and grown shaking at 37˚C. Exponential phase cells were harvested (~ 3hr) and β-galactosidase activity against an ortho-Nitrophenyl-β-galactoside substrate (ONPG) substrate was quantified as described elsewhere [123,124].

## Motility assays

Motility plates (0.3% agar) were prepared with M9 minimal medium with variable carbon sources (succinate, 30 mM; and maltose, 0.1 mM; glucose, 0.2%). Strains were grown overnight in LB medium and washed three times in M9 without a carbon source. Plates were inoculated via toothpick stabs and incubated 30˚C for 48-hr. The migration diameter (mm) was recorded.

## Bacterial two hybrid assays

Protein-protein interactions were detected using the BACTH bacterial two hybrid system [125]. *cheW* and *aerB* (excluding transmembrane domains and native start/stop codons) were cloned into SmaI-digested pUT18(C) (Kan$^R$) or pK(N)T25 (Amp$^R$) expression vectors to yield N-terminal T(18/25)-Aer or C-terminal CheW-T(18/25) fusions. Electrocompetent *E. coli* BTH101 were co-transformed with a pUT18 and pKT25 vector that carried either: an unfused adenylate cyclase domain (T18 or T25), the CheW-T(18/25) fusion or the T(18/25)-AerB fusion. Following 1 hour of outgrowth in SOC at 30˚C, 10 μL of concentrated outgrowth was spotted onto LB agar containing kanamycin and ampicillin (for selection), X-gal (for blue-white detection), and inducer (IPTG, 500 μM). Plates were incubated overnight at 30˚C and for an additional day at room temperature before being imaged.

## Anaerobic cultures

5 mL of M9 minimal medium without MgSO$_4$, CaCl$_2$, or carbon source were added to glass culture tubes. Tubes were sealed with rubber stoppers, crimped, purged for 10 cycles (20 sec vacuum, 20 sec N$_2$ purge), and autoclaved (gravity, 20 min). Post-autoclaving, the medium was amended with sterile solutions of MgSO$_4$ (to 2 mM), CaCl$_2$ (to 0.1 mM), glucose (to 0.5%) and with or without fumarate (to 50mM) using sterile syringes and needles. Tubes were injected with a *V. cholerae* cell suspension (1:100) and grown overnight shaking at 30˚C. Aerobic tubes containing M9 glucose (0.5%) with or without fumarate were included as a control. Congregation was measured via spectrophotometry, as described above.

## Supporting information

**S1 Fig. Targeted genetic mutations exclude the involvement of a variety of genes in *Δzur* congregation in M9 minimal medium.** (**A-H**) All strains were grown overnight in M9 minimal medium plus glucose (0.2%). All cultures were grown shaking (200 rpm), with the exception of static growth tested in panel (**A**). All cultures were grown in borosilicate glass tubes, with the exception of plastic tubes used in panel (**B**). Congregation was quantified by measuring the optical density (at 600 nm) of the culture supernatant before and after a brief vortex. The following mutants were tested in a *Δzur* background: (**C**) other putative Zur-regulatory targets (ABC-type transporter, *Δvca1098-vca1101;* che-III cluster, *Δvca1090-vca1097*), (**D**) chemotaxis genes (*cheA::STOP*, *ΔcheY*, *ΔcheZ*), (**F**) biofilm formation genes (*ΔvspL*), type IV pili (*ΔtcpA*, *ΔmshA*, *ΔpilA*, and *Δvc0502*), quorum sensing genes (*ΔcsqA*, *ΔcsqS*, *Δtdh*, *ΔluxS*, or *ΔluxQ*), (**G**) N16961 *hapR*$^{repaired}$, (**H**) the Vibrio Seventh Pandemic (VSP) island -I

(*Δvc0175-vc0185*), and regions of VSP-II (*"ΔVSP-II"*, *Δvc0491-vc0515*; *Δvc0490-vc0510*, *Δvc0511*, *Δvc0512*, *Δvc0513 or vc0513*::*STOP*, *Δvc0514 or vc0514*::*STOP*, *Δvc0515 or vc0515*:: *STOP*, or *Δvc0516*). (**E**) Congregation was also measured in a rough mutant (*vc0225*::*STOP*) and a rough mutant harboring deletions for *ΔfliC*, *ΔmotB*, or *Δvsp-II*. Data points represent biological replicates, error bars represent standard deviation, and asterisks denote statistical difference relative to the wild-type strain via (**A,B,G**) unpaired t-test or (**C-F, H**) Ordinary one-way ANOVA (****, $p < 0.0001$; ***, $p < 0.001$; *, $p < 0.05$).
(TIFF)

**S2 Fig. Transposon insertions that prevented *Δzur* from aggregating in M9 minimal medium.** (**A**) Table indicating the number of transposon insertions within motility, chemotaxis, and VSP-II genes for each of the screens (without pre-selection, v.1; with pre-selection of motile mutants, v.2) described in Fig 2. (**B**) Approximate location of transposon insertions (triangles) determined by arbitrary PCR [46] and Sanger sequencing are shown. (**C**) Strains carrying either an empty vector (-) or complementation vector (+) were grown overnight in LB medium with kanamycin. Strains were washed thrice with M9 minimal medium. A sterile toothpick was used to inoculate cells into M9 soft agar (0.3%) containing glucose (0.2%), kanamycin, and inducer (IPTG, 500 μM). The diameter of diffusion (mm) was measured following a 48-hr incubation at 30°C. Raw data points represent biological replicates, error bars represent standard deviation, and asterisks denote statistical difference via Ordinary one-way ANOVA test (****, $p < 0.0001$; **, $p < 0.01$).
(TIFF)

**S3 Fig. Annotation of the *verA* promoter region.** Diagram of the *verA* promoter region annotated with theh following features: predicted Zur box (red), predicted -10 and -35 regions (purple) [126], suggested start codon (**ATG**, green).
(TIFF)

**S4 Fig. Genes differentially expressed in *Δzur* relative to wild-type *V. cholerae* N16961.** (**A-B**) Volcano plots showing log 2-fold changes in gene expression in *Δzur* relative to wild-type; positive values represent up-regulation in *Δzur* and negative values represent down-regulation in *Δzur*. The y-axis denotes the negative log inverse of the p-value. Differentially expressed genes (log 2-fold change >1, adjusted p-value < 0.05) are denoted in red and are labeled with gene identifiers. Panel (**B**) shows the subset of genes within the blue box in Panel (**A**).
(TIFF)

**S5 Fig. Zur-dependent regulation of the *vca1098* promoter.** (**A**) Diagram of the *vca1098* promoter region annotated with the following features: predicted Zur box, red; predicted -10 region, purple [126]; transcription start site, +1 (5'-RACE); predicted ribosome binding site (RBS), yellow; proposed start codon (**ATG**), green. Asterisks indicate Zur box nucleotides (region "a" or "b") that were altered in the mutant reporters described below. (**B**) *vca1098* promoter *lacZ* transcriptional reporters (P$_{vca1098}$-*lacZ*, solid bars) or mutated versions (P$_{vca1098}$$^{\text{Zur box* a or b}}$-*lacZ*, striped bars) were inserted into a wild-type or *Δzur* background harboring a plasmid-borne, IPTG-inducible copy of *zur* (+) or empty vector control (-). Strains were grown overnight in LB and kanamycin, diluted 1:100 in fresh media containing inducer (IPTG, 400 μM), and grown for 3 hours at 37°C. Promoter activity (in Miller Units) was measured via β-galactosidase assays (See Methods and Materials). (**C**) Wild-type and *Δzur* strains carrying P$_{vca1098}$-*lacZ* or mutant derivatives were streaked onto M9 minimal medium agar with glucose (0.2%), X-gal, and with or without added zinc (ZnSO$_4$, 10 μM). Plates were incubated overnight at 30°C and then for an additional day at room temperature. *vca1098*

promoter activity is signified by a blue colony color.
(TIFF)

**S6 Fig. Construction of an *aerB* transcriptional reporter.** (**A**) Schematics for two attempted $P_{aerB}$-*lacZ* reporters containing either 400 bp or 1,314 bp of the promoter region are shown. (**B**) The $P_{aerB}$$^{400\ bp}$-*lacZ* and $P_{aerB}$$^{1,314\ bp}$-*lacZ* reporters were integrated into a wild-type or *Δzur* background and were struck onto LB X-gal plates. Plates were incubated overnight at 30˚C and then for an additional day at room temperature. $P_{aerB}$ expression is indicated by a blue colony color.
(TIFF)

**S7 Fig. AerB protein alignment with homologs from *V. cholerae*, *E. coli*, *A. brasilense, and S. oneidensis*.** (**A**) Results of protein BLAST [121] and (**B**) Clustal Omega alignment [127] of AerB (VC0512) with homologs from *V. cholerae* (AerA/VCA0658), *E. coli* (Aer/B3072), *A. brasiliensis* (AerC/AKM58_23950), and *S. oneidensis* (SO_1385). Conserved ligand binding and MCP residues targeted for mutation are indicated by orange and pink arrows, respectively.
(TIFF)

**S8 Fig. *V. cholerae* Δ*zur* congregation requires oxygen.** Wild-type and *Δ*zur were grown overnight in 5 mL M9 minimal medium plus glucose (0.5%) fermentatively (without a terminal electron acceptor) and cultured under aerobic (+ $O_2$) or anoxic (- $O_2$) conditions (see Methods for details). Tubes were grown shaking overnight at 30˚C and congregation was quantified via spectrophotometry as described previously. All data points represent biological replicates, error bars represent standard deviation, and asterisks denote statistical difference via Ordinary one-way ANOVA test (****, p < 0.0001; n.s., not significant).
(TIFF)

**S9 Fig. *V. cholerae* swarm assays with chemoreceptor mutants.** (**A-D**) The indicated strains were grown overnight in LB medium and washed thrice in M9 minimal medium lacking a carbon source. A sterile toothpick was used to inoculate cells into M9 soft agar (0.3%) with either (**A-B**) succinate (30 mM) or (**C-D**) maltose (0.1 mM) as a carbon source. The diameter of diffusion (mm) was measured following a 48-h incubation at 30˚C and representative swarms are shown (**A,C**). Note: Data for *Δzur*, *Δzur ΔaerB*, and *Δzur ΔaerA* are the same as shown in **Fig 5** and are shown here for comparison with a wild-type background. All data points represent biological replicates, error bars represent standard deviation, and asterisks denote statistical difference via Ordinary one-way ANOVA test (****, p < 0.0001; ***, p < 0.001, and n.s., not significant).
(TIFF)

**S10 Fig. Presence of VSP islands does not impact growth in zinc-chelated medium.** Wild-type, *Δvsp-I*, and *Δvsp-II* were grown overnight in M9 minimal medium with glucose (0.2%) at 30˚C. Cultures were washed twice and diluted 1:100 into (**A**) fresh M9 minimal medium glucose (0.2%), (**B**) plus the zinc-specific chelator TPEN (250 nM), or (**C**) plus TPEN and exogenous zinc (ZnSO$_4$, 1 μM). Growth at 30˚C of each 200-μl culture in a 100-well plate was monitored by optical density at 600 nm (OD$_{600}$) on a Bioscreen C plate reader (Growth Curves America).
(TIFF)

**S1 Movie. A *V. cholerae* N16961 Δ*zur* mutant congregates at the bottom of culture tubes.** *Δzur* was grown overnight shaking (200 rpm) in M9 minimal medium with glucose (0.2%) at 30˚C. The culture was manually agitated to disturb the pellet.
(MOV)

**S1 Table. Genes differentially expressed in *Δzur* relative to wild-type *V. cholerae* N16961.** Transcript abundances in *Δzur* relative to wild-type was measured using RNA-seq (see Methods for details). Gene ID's and putative ontology [82] are shown for all significant (adjusted p-value < 0.05) differentially expressed (log 2-fold change > 1) genes. Positive values represent up-regulation and negative values represent down-regulation in *Δzur* relative to the wild-type. Superscripts denote (a) a nearby canonical Zur box, (b) location on VSP-I or (c) location on VSP-II.
(XLSX)

**S2 Table. Genes differentially expressed in *a V. cholerae* strain overexpressing VerA.** Transcript abundances in a strain overexpressing VerA (VC0513) relative to an empty-vector control were measured using RNA-seq (see Methods for details). Gene ID's and descriptions [82] are shown for all significant (adjusted p-value < 0.05) differentially expressed (log 2-fold change > 1) genes.
(XLSX)

**S3 Table. Summary of strains and oligos used in this study.** Strains used in this study are listed with unique identifiers (SGM-#). For *E. coli* donor strains, the primers or gene block used to construct each plasmid are listed in the "Oligos" column. Genetic changes introduced into *V. cholerae* were screened using the method indicated in the "confirmation" column: either via PCR using the indicated primers (SM-#), via purification on kanamycin plates, or via blue-white screening on X-gal plates.
(XLSX)

## Acknowledgments

We thank Sean Murphy and Dr. Daniel Buckley for providing assistance with anoxic culturing, and Dr. John Mekalanos and Dr. Melanie Blokesch for generously sharing *V. cholerae* strains. We thank members of the Dörr lab for helpful discussions.

## Author Contributions

**Conceptualization:** Shannon G. Murphy, Tobias Dörr.

**Data curation:** Shannon G. Murphy.

**Formal analysis:** Shannon G. Murphy, Brianna A. Johnson.

**Funding acquisition:** Tobias Dörr.

**Investigation:** Shannon G. Murphy, Tobias Dörr.

**Methodology:** Shannon G. Murphy.

**Project administration:** Tobias Dörr.

**Supervision:** Shannon G. Murphy, Tobias Dörr.

**Validation:** Shannon G. Murphy, Brianna A. Johnson, Camille M. Ledoux.

**Visualization:** Shannon G. Murphy.

**Writing – original draft:** Shannon G. Murphy, Tobias Dörr.

**Writing – review & editing:** Shannon G. Murphy, Tobias Dörr.

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
