## [Decision Letter · Decision Letter 0]

14 Apr 2021

Dear Dr Doerr,

Thank you very much for submitting your Research Article entitled 'Vibrio cholerae’s mysterious Seventh Pandemic island (VSP-II) encodes novel Zur-regulated zinc starvation genes involved in chemotaxis and autoaggregation' to PLOS Genetics.

The manuscript was fully evaluated at the editorial level and by independent peer reviewers. The reviewers appreciated the attention to an important topic but identified some concerns that we ask you address in a revised manuscript

We therefore ask you to modify the manuscript according to the review recommendations. Your revisions should address the specific points made by each reviewer.

[LINK]

Yours sincerely,

Diarmaid Hughes

Associate Editor

PLOS Genetics

Josep Casadesús

Section Editor: Prokaryotic Genetics

PLOS Genetics

Reviewer's Responses to Questions

**Comments to the Authors:**

Reviewer #1: The manuscript by Murphy et al. identifies a novel role for the second Vibrio pathogenicity island (VSP-II) in the physiology of toxigenic Vibrio cholerae. Specifically, they identified that V. cholerae ∆zur mutants aggregate out of solution when grown aerobically in minimal medium, and then conduct a number of genetic screens which led them identify a regulatory module on VSP-II that was critical for this response. This regulatory module is composed of a transcriptional activator, VerA that induces the expression of a chemotaxis gene, AerB. Using carefully controlled genetic experiments and chemotaxis assays they nicely demonstrate that AerB is sufficient to induce this response and likely promotes a negative chemotactic response to oxygen. The importance of this physiological response during the V. cholerae lifecycle remains unclear, but the authors speculate on this point in the Discussion of the manuscript. Overall, this study was well conducted and very clearly written. The findings represent an important advance to our understanding of V. cholerae biology – especially because it identifies the function of genes associated with one of the genomic islands that distinguish isolates from the current pandemic. I have only minor points for the authors to consider. Namely, to speculate on and or test further the mechanism underlying bacterial aggregation during this response.

Based on the data presented in Fig. 1C it is assumed that the WT grown without ZnSO4 does not experience Zinc starvation. Is this expected due to the efficiency of zinc uptake and the trace amounts of zinc available in the medium? Data in Figure 3 using their PverA reporter construct suggest that this may be the case, but the authors should formally comment on this point earlier in the text.

Line 142 – “S1F” not “SF”

Figure S1F – does the ∆type IV pili strain really have VC0504 mutated or is it supposed to read VC0502? If the former, an explanation might be warranted since this is not a putative major pilin gene.

It is striking that verA was hit so many times in the screen, while aerB, which is sufficient to carry out the phenotype, was only hit once. Can the authors speculate on why this may have occurred? It is not essential to test this experimentally, but do they know if aerB mutants were poorly represented in their input Tn mutant pool?

Line 160 and 180 – Define the “vc0513-vc0515 promoter region” as “PverA” earlier in the manuscript, since this term is already used.

Line 163-164 - It is unclear how introduction of a premature stop codon would be more likely to prevent polar effects on downstream genes because Rho-dependent termination could, in fact, still influence the transcription of downstream genes. An in-frame deletion on the other hand would be less likely to suffer from this shortfall. The authors nicely demonstrate that the genes can be complemented in trans, so I am not questioning the results obtained in any way. But I would suggest that they remove this general statement about stop codons unless they can provide additional supporting evidence for it.

Line 213 – should refer to “S1B”

Line 241 – refer to Fig. 4B at the end of this sentence.

Line 319 – “taxis” not “axis”

Line 1100 – vca0685 is an ABC transporter component, not an MCP. Should be vca0658 according to the Methods section.

Figure S7 - The TPEN concentrations used don’t seem to generate zinc limiting conditions for V. cholerae growth since growth kinetics and final growth yield appear unchanged compared to growth when TPEN is omitted. Would higher doses of TPEN reveal a difference in growth? Even if there is not a difference in optical density, could there be a difference in viability that is revealed by quantitative plating? This may fit with the authors hypothesis that this response is required for optimal resistance to redox stress.

It is striking that aggregation and cell pelleting is the end point phenotype observed if AerB simply promotes a negative chemotactic response to oxygen. If this was simply chemotaxis away from oxygen, I would expect cells to simply swim to the lower part of the culture tube, not actually aggregate out of solution. Is aggregation actually occurring through cell-cell interactions promoting the formation of bacterial clumps? This could be easily assessed by microscopy. If bacteria do form clumps, can the authors speculate on what is promoting cell-cell interactions? Their results nicely demonstrate that the canonical factors thought to mediate cell-cell interactions in V. cholerae are not playing a role here. Since motility seems to play a role in this response could it be tangling of the polar flagellum between cells?

Is the oxygen-dependent aggregation response reversible? For example, if ∆Zur or AerB overexpressing cells are grown aerobically and then transitioned to anaerobic conditions, do bacterial aggregate dissipate? If AerB is directly sensing oxygen to modulate motility as proposed, I would expect this response to be reversible.

AerA seems to carry out the opposite phenotype to AerB. Do ∆zur ∆aerA mutants aggregate more quickly than ∆zur mutants? This might be expected if the aerotaxis response is sufficient to promote aggregation.

It may be worth mentioning in the Discussion that AerB could simply be upstream of additional genes required for this response. The authors nicely demonstrate that ectopic expression of AerB recapitulates the aggregation phenotype even when Zur is intact. In the future, they may want to consider carrying out a genetic selection in the AerB overexpressing strain. Since aerB was poorly hit in their original screen, this modified screen setup may help uncover additional genes that are required for this response.

Reviewer #2: I really enjoyed reading this manuscript “Vibrio cholerae’s mysterious Seventh Pandemic island (VSP-II) encodes novel Zur regulated zinc starvation genes involved in chemotaxis and autoaggregation”. In this work, the authors examined an aggregation phenotype in V. cholerae Zur mutants, which mimic a zinc starvation response. They characterize the downstream pathway leading to this novel zinc-starvation response including induction of a transcriptional regulator, VerA, within VSP-II. They further elaborate on the VerA regulon using RNA-Seq to show that it induces expression of its own operon as well as the neighbouring gene, AerB. AerB instigates energy taxis away from oxygen, resulting in the observed aggregation phenotype.

Overall the experiments are well conducted, have appropriate controls and supporting data, and are clearly presented. While the biological role of this phenotype and the in vivo environment where it is relevant remain unknown, the results presented support the conclusions made. These findings provide a significant step forward to the field by describing a novel genetic pathway, phenotype, and energy taxis mechanism in V. cholerae, with potential links to virulence.

1) I am still intrigued by what the biological role of this pathway and phenotype could be. The authors noticed the aggregation phenotype using a zur mutant but did not find a condition that could induce similar phenotype in wild type. Could it be that there is environmental Zn or that Zur responds to other metals?

2) Regarding the aggregation phenotype, the data presented supports swimming away from oxygen, but the modest effect in the succinate swarming assay does not look strong enough to hold the cells together upon vortexing. It seems that there must be something sticking the cells together in an aggregate. You already tested known pili, vps etc., and your Zur RNA-Seq suggests VPS/biofilm are also downregulated. Could any of the other hits in the RNA-Seq of the Zur mutants be possible candidates? Were any of these knocked out to test aggregation?

3) The zur mutant was found to aggregates at the bottom of the tube after overnight incubation with shaking. One would think that the oxygen gradient in the shaking tube is not simply from top aerial-liquid interface to tube bottom. How could this bottom aggregation in shaking tubes be driven by AerB-mediated oxygen sensing?

4) Maybe it is my personal preference but please do not use the subjective term “mysterious” in the title and text.

5) Line 246: “aerB is part of a much longer transcript (extending >1 kb upstream of the start codon).” Did you test for any potential upstream or cryptic ORFs in this upstream region? Could these play a role in the aggregation phenotype or the regulation of aerB?

6) Line 212-213: “Neither the transporter nor the che-3 cluster, however, were required for Δzur aggregation (Fig. S1F)." Figure S1F uses gene names while Fig. S3 and the previous text use VC numbers. This makes it difficult to verify which genes are referred to in this sentence. Please edit this sentence or the earlier one to clarify which genes are testing the transporter and the che-3 cluster in Fig. S1F.

7) Line 339: missing the word ‘is’, “ the human host is a well-studied…”

8) There are a few typos in calling figures. Also, in numerous instances, figures or panels are called out of order. I personally don’t mind this, especially for supplemental figures when it puts similar panels together, but some other readers might find it distracting. If the supplemental figures/panels can be easily rearranged it could be worthwhile. I may have missed others as well, but some examples listed below:

a. Line 283: Fig. S5 appears after S6

b. Line 142: Fig. SF -> Fig. S1F. Also, this appears before S1E

c. Line 290: Fig. 6A -> Fig. 5A; again out of order

Reviewer #3: This is an excellent paper that relies on clever genetic screens, gene expression studies and phenotypic assays to uncover the mechanistic basis for aggregation of V. cholerae lacking Zur, a key regulator of the response to zinc starvation. These multi-faceted studies lead to a highly plausible model to explain the aggregation phenotype. The paper will be of considerable interest to PLOS Genetics readers interested in bacterial genetics and physiology. One short coming of the work, which the authors acknowledge and that should not delay publication, is the absence of experiments to decipher when this mechanism is important for the pathogen. The authors frame the work suggesting that these studies might advance understanding of the predominance of the El Tor biotype, but they don’t deliver on that suggestion.

Major points:

1. Figure 3 should include a MA plot and/or a volcano plot showing all genes and annotated with genes the authors want to highlight; e.g., the authors allude to increased expression of ctxAB and Tcp proteins in the zur deficient background but the reader has no idea of the magnitude of these changes. In Figure 3D, the heat map scale should include numbers.

2. While it is difficult to discern when VerA/AerB dependent mechanisms confer fitness to the pathogen, there is simple (non-animal) experiment that might provide a clue- qPCR or RNA-seq analysis in El Tor virulence inducing conditions (AKI media).

3. Fig 5D is not particularly convincing. Would overexpression of AerB have the opposite effect?

Minor points:

1. classical biotype is not capitalized.

2. Add to discussion recent evidence that VSP-encoded ORFs may be phage defense elements (https://www.biorxiv.org/content/10.1101/2021.03.31.437871v1 ). Some of these enzymes are actually Zn-dependent - would be nice to see some contextualization/discussion of how these findings relate to each other (if at all).

3. Line 270 change ‘indicate’ to ‘strongly suggests’ [these data are not definitive]

4. Line 290 incorrectly refers to Fig 6A

5. Reference 1 is out of date. Lancet published a new cholera review since then.

6. Is VerA or AerB under-represented in the in vivo TnSeq publications?

7. Additional discussion/exploration of the relationship of AerA and AerB would be useful; e.g. are they expressed at the same time?

8. Does classical delta zur aggregate?

**Have all data underlying the figures and results presented in the manuscript been provided?**

Reviewer #1: Yes

Reviewer #2: Yes

Reviewer #3: Yes

PLOS authors have the option to publish the peer review history of their article (what does this mean?). If published, this will include your full peer review and any attached files.

Reviewer #1: No

Reviewer #2: No

Reviewer #3: **Yes: **Matthew Waldor

---

## [Decision Letter · Decision Letter 1]

27 May 2021

Dear Dr Doerr,

We are pleased to inform you that your manuscript entitled "Vibrio cholerae’s mysterious Seventh Pandemic island (VSP-II) encodes novel Zur-regulated zinc starvation genes involved in chemotaxis and cell congregation" has been editorially accepted for publication in PLOS Genetics. Congratulations!

Yours sincerely,

Diarmaid Hughes

Associate Editor

PLOS Genetics

Josep Casadesús

Section Editor: Prokaryotic Genetics

PLOS Genetics

Comments from the reviewers (if applicable):

Reviewer's Responses to Questions

**Comments to the Authors:**

Reviewer #1: The authors have reasonably addressed the comments and questions I raised in the initial round of review and I believe that the manuscript is now suitable for publication.

One thing the authors may want to consider testing in the future is the reversibility and speed of this congregative response, which might help elucidate how directly / indirectly aerB facilitates this process. This could be tested in two ways: First, by growing a strain ectopically expressing aerB aerobically and then adding oxyrase to this culture to induce an anaerobic environment (to see if this dissipates congregation and also to determine the timescale in which it occurs). And second, by taking cells ectopically expressing aerB and growing them anaerobically, followed by exposure of this culture to air (to see if this induces congregation and also to determine the timescale in which it occurs). If aerB is promoting congregation purely through a chemotactic response, then these experiments should reveal a rapid reversal of phenotype. Furthermore, it would be expected that inhibition of transcription and/or translation (through use of appropriate antibiotics) should not influence these transitions.

Reviewer #2: I am pleased with all the revisions the authors have made.

Reviewer #3: this is a cool paper that will be of interest to PloS Genetics readers

**Have all data underlying the figures and results presented in the manuscript been provided?**

Reviewer #1: Yes

Reviewer #2: Yes

Reviewer #3: Yes

PLOS authors have the option to publish the peer review history of their article (what does this mean?). If published, this will include your full peer review and any attached files.

Reviewer #1: No

Reviewer #2: No

Reviewer #3: No

**Data Deposition**

http://datadryad.org/submit?journalID=pgenetics&manu=PGENETICS-D-21-00391R1

**Press Queries**

---

## [Editor Report · Acceptance letter]

16 Jun 2021

PGENETICS-D-21-00391R1 

*Vibrio cholerae*’s mysterious Seventh Pandemic island (VSP-II) encodes novel Zur-regulated zinc starvation genes involved in chemotaxis and cell congregation 

Dear Dr Dörr, 

We are pleased to inform you that your manuscript entitled "*Vibrio cholerae*’s mysterious Seventh Pandemic island (VSP-II) encodes novel Zur-regulated zinc starvation genes involved in chemotaxis and cell congregation" has been formally accepted for publication in PLOS Genetics! Your manuscript is now with our production department and you will be notified of the publication date in due course.

With kind regards,

Katalin Szabo

PLOS Genetics

On behalf of:
